# Identification of core therapeutic targets for Monkeypox virus and repurposing potential of drugs: A WEB prediction approach

Huaichuan Duan[1⊙], Quanshan Shi[2⊙], Xinru Yue[2], Zelan Zhang[2], Ling Liu[2], Yueteng Wang[2], Yujie Cao[2], Zuoxin Ou[2], Li Liang[2], Jianping Hu[2]*, Hubing Shi[1]*

1 Laboratory of Tumor Targeted and Immune Therapy, Clinical Research Center for Breast, State Key Laboratory of Biotherapy, West China Hospital, Sichuan University and Collaborative Innovation Center for Biotherapy, Chengdu, China, 2 Key Laboratory of Medicinal and Edible Plants Resources Development of Sichuan Education Department, School of Pharmacy, Chengdu University, Chengdu, China

⊙ These authors contributed equally to this work.
* shihb@scu.edu.cn (HS); hjpcdu@163.com (JH)

**Data Availability Statement:** All relevant data are within the paper and its Supporting information files.

## Abstract

A new round of monkeypox virus has emerged in the United Kingdom since July 2022 and rapidly swept the world. Currently, despite numerous research groups are studying this virus and seeking effective treatments, the information on the open reading frame, inhibitors, and potential targets of monkeypox has not been updated in time, and the comprehension of monkeypox target ligand interactions remains a key challenge. Here, we first summarized and improved the open reading frame information of monkeypox, constructed the monkeypox inhibitor library and potential targets library by database research as well as literature search, combined with advanced protein modeling technologies (Sequence-based and AI algorithms-based homology modeling). In addition, we build monkeypox virus Docking Server, a web server to predict the binding mode between targets and substrate. The open reading frame information, monkeypox inhibitor library, and monkeypox potential targets library are used as the initial files for server docking, providing free interactive tools for predicting ligand interactions of monkeypox targets, potential drug screening, and potential targets search. In addition, the update of the three databases can also effectively promote the study of monkeypox drug inhibition mechanism and provide theoretical guidance for the development of drugs for monkeypox.

## 1. Introduction

On July 23, 2022, WHO declared the Monkeypox virus (MPV) outbreak a "public health emergency of international concern", marking the beginning of the global war. From the first explosion in the United Kingdom in May, monkeypox rapidly spread to 110 countries, with a cumulative number of 85860 reported infections and 93 deaths [1]. Monkeypox is a viral zoonotic disease caused by MPV of the genus orthopoxvirus. In 1958, monkeypox was first identified in experimental macaques and the first human infection occurred in 1970 [2, 3]. From a

**Funding:** The National Key Research and Development Program of China (No. 2022YFD2200101-04). The funders had no role in study design, data collection and analysis, decision to publish, or preparation of the manuscript.

**Competing interests:** The authors have declared that no competing interests exist.

long time, monkeypox was endemic only in countries like central and west Africa and is transmitted from animal to human mainly through direct contact with the blood, body fluids, skin or mucous membrane damage of infected animals. As a self-limiting disease, the symptoms of monkeypox usually last for 2 to 4 weeks and are mainly manifested as fever, myalgia, swollen tonsils, headache, systemic papulae, etc., with many complications including secondary infection, bronchopneumonia, sepsis, encephalitis and corneal infection leading to vision loss [4–6].

MPV is an enveloped double-stranded DNA virus with a length of 197kb, mainly divided into Central African and West African evolutionary clades [7]. Of these, the Central African generally causes more severe disease, with higher case fatality and more frequent transmission rates, while the West African, the current spreading clade, is milder. The morphology of MPV is oval with a size about 200–400 nm, the surface is a lipoprotein membrane, while the middle is composed of two protein-containing side bodies and a thick membrane core containing a double-stranded DNA [8, 9]. In the replication process of MPV, there are two modes: mature virion (MV) and extracellular enveloped virion (EV) (Fig 1). EV has an envelope from the membrane of the Golgi apparatus, which breaks down after contact with the cell, exposing the outer membrane and then directly fusing with the cell membrane; while MV can enter the cell directly by fusing with the cell membrane or endocytosis, releasing the enveloping viral core. The site of viral replication is the endoplasmic reticulum, where the virions gradually mature and form infectious MV after a series of identification, processing and synthesis. Thereafter,

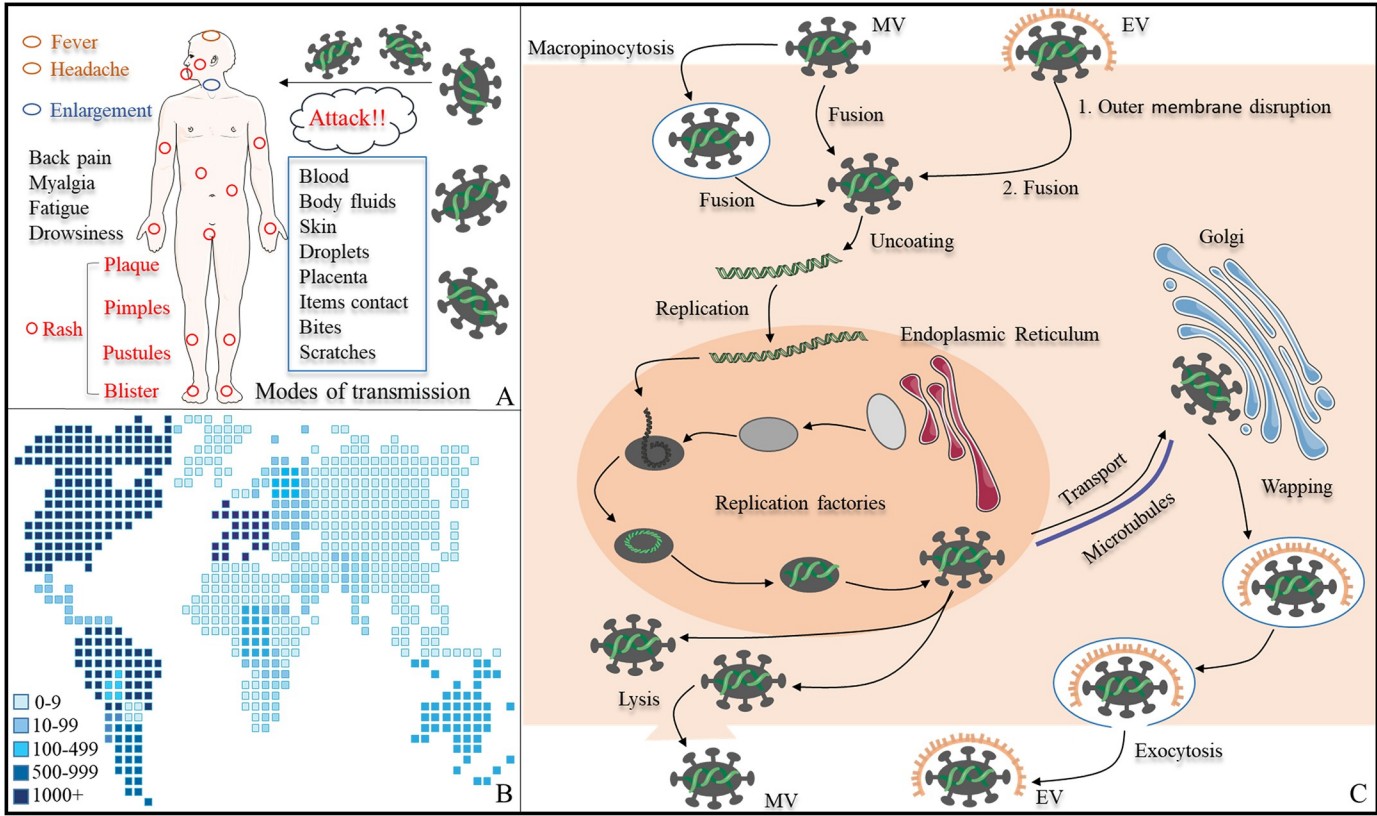

**Fig 1. Distribution and replication of monkeypox.** (A) Infection mode and common symptoms monkeypox; (B) cumulative distribution of confirmed monkeypox cases worldwide by region; (C) cycle of infection, replication, and release of monkeypox.

mature MV can directly release outside the cell through cell lysis. In addition, it can also be secreted outside the cell through the Golgi apparatus by the cell's secretory transport system. In this process, the virus will obtain an extra layer of membrane and finally form EV [10–12].

Since the 1970s, the treatment of monkeypox has been a major scientific problem in life science, which mainly focuses on symptomatic supportive treatment to prevent complications, while antiviral drugs such as sidofovir are only used for the treatment of severely infected people due to their obvious toxic and side effects. Therefore, the vaccine becomes the most effective barrier against the disease, and it is worth mentioning that smallpox vaccination effectively makes about 85% of the population immune to the MPV. With the recurrence of monkeypox in recent years, the research on monkeypox has gradually increased. In 2019, a modified attenuated non-replicative cowpox virus (Ankara strain) vaccine Jynneos (also known as Imvamune or Imvanex) was approved for the prevention of MPV, and the smallpox vaccine Acam2000 was approved for the prevention of MPV [13]. In 2022, Dr. João provided the monkeypox genome sequence using DNA extraction and shotgun metagenomics sequencing techniques, laying the foundation for further understanding of its epidemiology, sources of infection, modes of transmission and vaccine development [14]. In October of the same year, Wuhan Institute of Biological Products in China successfully isolated MPV strains from clinical samples of patients, and further started the research on vaccines, drugs and other related work, which greatly promoted the monkeypox prevention and treatment project (https://www.medicoswab.com/china-has-successfully-isolated-monkeypox-virus-strain-and-started-vaccine-drug-research/). As the first barrier to MPV, vaccines can effectively prevent the occurrence of the disease and cure patients with mild cases, but not for patients with deep infection [15]. In recent years, scientists have conducted a lot of research to obtain effective anti- monkeypox virus drugs through bioinformatics analysis, drug database screening, homologous disease drug modification and other methods [4–6, 8, 9, 16]. In 2013, Smee et al. obtained vaccinia immune globulin, cidofovir, ST-246 and CMX001 with superior activity by summarizing the poxvirus inhibitors before 2012 and screening them through animal infection assays [17]. Based on broad-spectrum anti-DNA viral drugs, Rizk et al. found that tecovirimat, cidofovir, and brincidofovir showed promising activity against monkeypox in vitro and experimental models. And brinsidovir was approved by FDA in 2021 for the treatment of smallpox [18, 19]. As one of the first FDA drugs for smallpox, tecovirimat was approved by EMEA (the European Medicine Agency) in 2022 for the treatment of smallpox, monkeypox, cowpox, and pox complications by binding to viral genes and preventing the release of the virus from cells [20–23]. The lack of an effective drug for monkeypox and the limitations of existing vaccines have led to an unpredictable global epidemic that poses a great threat to human health and survival and a devastating blow to global economic recovery. In addition, the lack of a standard treatment for monkeypox worldwide is still a critical situation, and it's urgent to develop a specific drug that can be used as a therapeutic.

At present, there are two main ideas for the treatment of monkeypox in the world: 1. Based on the monkeypox structure design, including the search for lead compounds, drug synthesis intermediates or metabolites to obtain drugs that can effectively combat the virus. But the whole process is time consuming, labor-intensive, consumable materials, low success rate, cannot cope with the current urgent situation; 2. Apply old drugs to new ones. Potential inhibitors are obtained by screening the activity of a large number of identified drugs or small molecule compounds, which greatly shortens the drug discovery process [24]. However, how to screen compounds quickly becomes another problem? In recent years, the proliferation of online platforms has greatly facilitated the response to this problem. In 2021, Daniele collected the genetic sequences of SARS-CoV-2 virus through multiple channels and developed a software that can track the mutation status of new coronaviruses in real time and provide users with

annotations of viral genome sequences: coronapp. Notably, the development of coronapp helps to monitor genomic differences in SARS-CoV-2 in a timely manner, visualize the current SARS-CoV-2 mutation status globally, and provide possible directions for novel new SARS-CoV-2 drugs [25]. In addition, understanding pathogen transmission and evolution is important for effective public health measures and surveillance. In 2018, James et al. used a database of viral genomes, a bioinformatics pipeline for phylodynamics analysis, and an interactive visualisation platform to build a real-time website showcasing the evolution and spread of viral pathogens, Nextstrain. It can monitor virus evolution from a phylogenetic perspective and provide visual epidemiological data such as the number of cases and deaths [26]. In drug design, the interaction between target ligands is a key factor for molecular recognition. Chang et al. built a web server that can be used to predict the binding mode of the COVID-19's target with ligands (small molecules, peptides and antibodies) by collecting and constructing the protein structure related to the life cycle: the COVID-19 Docking Server, which effectively promotes the development of neo-crown drugs [27]. However, there are still no online tools that can systematically provide bioinformatics information on monkeypox, possible drug targets, and potential drug inhibitor structures, nor are there any tools dedicated to a server for predicting the binding pattern of monkeypox targets to ligands.

To overcome these limitations, we have developed Monkeypox virus Docking Server (http://monkeypox.hulab.org.cn/), a web application with three purposes: Provide information on key targets of monkeypox and possible modeling structures, construct a library of potential drug or inhibitor, and provides a free and interactive tool for the prediction of monkeypox target-ligand interactions and following drug discovery for monkeypox. Through the web-server, people can facilitate to evaluate the binding modes and binding affinities between the targets and small molecules. By launching the server, hoping to provide a free and easy to use tool for people who is interested in drug discovery against monkeypox.

## 2. Methods

### 2.1 Open Reading Frame (ORF)

ORF is a fundamental concept in molecular bioinformatics, where open refers to the "open" region of an intact gene used for protein translation, and reading frame refers to one of the six possibilities for translation of a double-stranded gene sequence into amino acids. In molecular biology, the detection of ORF is an important step in the discovery of specific protein-coding genes in the genome sequence. In our work, the acquisition of ORF mainly includes literature reading and database research.

### 2.2 Collection of targeted protein sequence

The amino acid sequence of the targeted protein on orthopoxvirus including variola (VAR), vaccinia (VAC), cowpox (CPV) and monkey virus was collected from the National Centre for Biotechnological Information (NCBI) database. Accurate protein sequences are the key to affinity analysis and the prerequisite to ensure the accuracy of the modeled structure.

### 2.3 Advanced protein modeling

The three-dimensional structure of proteins is an important basis for understanding the biological functions and for structure-based drug design. With the rapid development of structural biology, the speed of measuring protein 3D structures using methods such as NMR or X-Ray crystallography has been greatly improved. However, the 3D structures of some proteins are still difficult to determine due to many reasons, such as excessive molecular weight or

difficulty in crystallization. Such scenarios necessitate the use of computational simulations to construct the three-dimensional structures of proteins. Among the many computer modeling methods, Homology modeling is widely used as the most mature, accurate and reliable means of results. In our work, two pairs of methods are used to perform modeling based on different states of proteins.

**2.3.1 Sequence-based homology modeling.** Discovery Studio is a comprehensive molecular simulation software that serves scientific researchers in the life sciences and drug design fields, allowing the design and property prediction of protein modifications starting from the protein structure. DS-based modeling is done by giving an unknown structural sequence and another high homology protein with a known 3D structure [28]. Through the Modeler and Sequence Analysis modules, protein structures can be automatically and rapidly predicted. It is easy to see that this method can only be used for systems with high homology and crystal structure, and is no longer applicable for systems without crystal structure information.

**2.3.2 AI algorithms-based homology modeling.** AlphaFold 2 is a deep learning model for predicting the 3D spatial structure of proteins. In this algorithm, we only need to give the protein sequence to get the structure prediction with "atomic accuracy" level [29–32]. It is worth mentioning that this method achieves template-free modeling through four main points: training the neural network to perform Iterative refinement on the regression target; extensive use of the attention architecture to reduce the sample complexity; using BERT-like masking operation for various input Information plus noise requires stable output; multi-label data training to enhance the rationality of modeling. In summary, based on the above 2 methods, for all proteins with complete sequences, we can reasonably and effectively mold the 3D structure.

## 2.4 Webtool of MPV

**2.4.1 Server construction.** Monkeypox virus docking server is based on CentOS for Linux, the website application is deployed using Docker containerization technology, and the website requests on port 80 are proxied and forwarded through NGINX [33]. The front-end of the site uses the CSS framework Bootstrap grid technology with its forms, buttons, tables and other controls for layout. In addition the use of animate CSS to enhance the dynamic effect of image files, Javascript technology with the Jquery framework, toast and other plug-ins to enhance page interaction [34]. The use of revolvermaps control for site access source records. The back-end of the website uses PHP framework and the overall design is carried out by ThinkPHP. Use single-entry mode to take over all requests, in addition, MVC (Model+View +Controller) design pattern first by Controller to get the request information, Model to query the database information, View to parse the template and generate the front-end interface, and finally by Controller to return the response [35, 36].

After user submit a job, the user's task information is written to Mysql by the PHP/ PDO extension, and the task is in the pending execution state [37]. Use the Crontab command to call the written PHP command handler, this handler is divided into task execution and task result checking [38]. Then is the processing of user input files, through the Obabel program to unify the user's file processing to mol2 format, and then according to the user to fill in the information to generate the configuration file. According to the protein classification selected by the user to obtain the corresponding structure file, call the AutoDockTools program generated by the pdbqt format file [39]. Finally, the vina program is called for docking, and the task is in execution [40]. Vina program generates the corresponding out and log files after running. The PHP program parses the scoring results and splits the results of the 10 models with vina_split and uses the zip command in linux to generate a zip file for download.

**2.4.2 Implementation.** The server page is divided into Home, Submit, Check Result, Job Status, Example and Support Information. Home page from top to bottom is "rotating image display", "about us", "our features" and "advantages", Submit page contains two functions: small molecule docking and protein docking. In the small molecule docking module, users need to select the protein target, docking mode (single molecule or batch docking), upload small molecule files (mol2, smi, sdf), Exhaustiveness configuration and provide email address for receiving task completion email. For protein docking, users need to upload protein files, select box center coordinates, box size, Exhaustiveness configuration, small molecule database selection and email address. The Job Status page shows all tasks currently running on the site (emails and task IDs have been made private), and the "Example page" shows a sample of a successful task, including scoring and ranking information, results file download, and variable molecule view.

## 3. Results

### 3.1 Current status of monkeypox data

ORFs are a fundamental concept in molecular biology and bioinformatics, and their detection is an important step in the discovery of specific protein-coding genes in the genome sequence. As typical orthopox viruses, we collected a total of 191 ORFs of monkeypox (residue > 60) and summarized the potential functions of each (S1 Table). Then, genetic comparisons with VAR, VRC and CPV revealed that genes essential in other orthopox viruses are present in MPV, notably close to 95% of ORFs have sequence identity (sequence similarity > 85%) and the central region of the genome where D4L to A26R are highly similar. By genetic identity, we can infer potential targets in MPV by orthopox virus. Back in 2004, Byrd et al. co-cultured TTP-6171 with a variety of poxviruses through a 24-hour exposure with almost undetectable toxicity. They showed that TTP-6171 could effectively affect the I7L gene in various pox viruses, including MPV and smallpox, to achieve therapeutic effects [41]. In 2021, Park et al. found that MPV, like VAC, manifested inhibitory effects and was closely related to the amount of dsRNA produced by a series of C3L homologs of orthopox viruses interacting with mammalian PKR [42]. In this year, Li et al. comparatively analyzed the binding pattern between C19L of VAR and MPV and tecovirimat based on molecular docking and determined that C19L plays a crucial role in the formation and infectivity of EV [43]. The approach of seeking proteins based on viral gene consistency has been further validated in ongoing experiments, so in our work, we have obtained potential key proteins from extensive literature analysis for MPV and orthopox viruses.

### 3.2 MPV potential targets protein

During MPV infection, a large number of proteins are involved in the process of virus invasion, replication, envelopment and release, and MPV can be effectively suppressed by inhibiting these key proteins. In recent years, scientists have conducted numerous studies on various processes of infection and finally identified 12 key proteins that have important roles for the virus (Table 1).

**3.2.1 A49R.** Virus-encoded nucleoside kinase homologs are thought to be precursors for viral DNA synthesis, and unlike acyclic nucleoside phosphates, nucleoside analogs usually require phosphorylation to triphosphate to have antiviral activity [44]. A49R: a thymidine kinase that catalyzes the phosphorylation of thymidine monophosphate to diphosphate and also phosphorylates 5'halo-deoxyuridine monophosphate analogs. It is worth mentioning that A49R differs significantly from human thymidine kinase in active site and structure, and A49R

**Table 1. Isologs, function and residue information of 12 key proteins in MPV replication.**

| ORFs | Isologs | Function | Key residues |
|---|---|---|---|
| A49R | VAR/J2L, VAC/A48R, CPV/A51R | Thymidine kinase | R41, D13, Y144 |
| C3L | VAR/C3L, VAC/K3L, CPV/M3L | Eukaryotic translation initiation factor-2a homolog protein | A25, Y27, I39 |
| H1L | VAR/I1L, VAC/H1L | Tyrosine/serine protein phosphatase, blocks IFN-γ signal | N115, G113, V114 |
| E4R | VAR/F4R, VAC/D4R | Uracil DNA glycosylase | G215, F216,Y218 |
| C7L | VAR/C5L, VAC/F1L, CPV/G1L | Hypothrtical protein | V115, L158, G155 |
| A41L | VAR/A46L, VAC/A41L, CPV/A43L | Secreted protein reduces influx of inflammatory cells | F21, L47, F181 |
| D6L | VAR/D5L, CPV/C8L | Interleukin 18-binding protein | Y53, L55, F67 |
| E13L | VAR/N3L, VAC/D13L | Needed for immature IMV surface membrane | F168, F486,F487 |
| A22R | VAR/A21R, VAC/A20R | Processivity factor for viral DNA pol | Q146, G203, E204 |
| I7L | VAR/K7L, VAC/I7L | Virion core protein, DNA topoisomeras II homolog | L239, H241, C328 |
| C19L | VAR/C17L, VAC/F13L, CPV/G13L | Envelope protein | C53, N55, S58 |
| A50R | VAR/J4R, VAC/A50R, CPV/A53R | DNA ligase | C11, Y15, C147 |

becomes an attractive research target for the development of promising antiviral chemotherapeutic drugs [45, 46].

**3.2.2 C3L.** The antiviral protein kinase R (PKR) is an important host restriction factor, which poxviruses must overcome to productively infect host cells [47]. During the invasion of monkeypox into the host, in 2002, Dar found that viral inhibition of PKR was diminished by mutating a conserved region in C3L. He thought C3L could produce homologs of elF2α that competitively bind to PKR and inhibit dimer formation, which in turn promotes viral infection [48]. In this year, based on the specific inhibition of PKR exhibited by a series of C3L direct homologs, Park et al. proposed that the sensitivity of host proteins towards inhibition cannot be inferred from phylogenetic correlations alone, but must be determined experimentally. This also gives an indication for drug design against C3L: drug design must be experimentally based, not homologous substitution [42, 49].

**3.2.3 H1L.** Membrane fusion is the first step in viral invasion of the host, and in parallel to transporting the viral core into the cytoplasm, two supporting protein lateral bodies (LBs) are required for translocation [50]. LBs are mainly composed of phosphorylated protein C23R, bispecific phosphatase H1L and oxidoreductase C10L, of which H1L can participate in STAT1 dephosphorylation to prevent interferon G-mediated antiviral responses and is the immunomodulatory activity of key protease for immunomodulatory activity [51–53]. By inhibiting the process of phosphorylation, the immune response of the host can be effectively promoted, thus greatly enhancing the anti-infective ability of the organism [54].

**3.2.4 E4R.** The proteins involved in the viral replication process have been used as key targets for drug design [55]. In MPV replication, E4R, a highly conserved protein, is an effective therapeutic target for blocking viral replication. Functionally, it has dual functions of DNA repair (as a uracil-DNA glycosylase) and processed DNA synthesis [56–58]. At the same time, it binds to A23R, allowing DNA polymerase and DNA to associate tightly and not separate from DNA while synthesizing the extension strand [59]. Back in 2016, Nuth et al. identified

G215-I217 as an essential region for the function by analyzing the structure of E4R, providing a direction for subsequent drug design [60].

**3.2.5 C7L.** Subversion host apoptosis is a critical survival strategy for viruses to secure both their proliferation and survival [61]. Certain viruses can prevent rapidly clearing infected host cells by exhibiting sequence, structural and functional homology to the mammalian pro-survival B-cell lymphoma 2 (Bcl-2) protein. In monkeypox, virulence factor C7L is a potent inhibitor of apoptosis and acts by binding to the pro-apoptotic Bim [62, 63]. 2015, Marshall et al. performed biochemical analysis and functional characterization of C7L and found that the inhibition mechanism of apoptosis was not identical among different orthopox viruses, but its structure and function were basically consistent [64].

**3.2.6 A41L.** In addition to viral replication, viral infection is also quite an important process. A41L, a glycoprotein with biological properties consistent with chemokines, is a viral CC chemokine inhibitor (vCCI) [65]. Structurally, the protein is similar to the vCCI protein family, but the surface loops and electrostatic charge distribution differ significantly. The biological data revealed that A41L facilitates virus recognition by competitively interacting with glycosaminoglycans and occupying chemokine binding sites. In summary, the design of potential inhibitors against A41L could prevent infection starting from virus recognition [65–68].

**3.2.7 D6L.** Human interleukin-18 (hIL-18) plays an invaluable role in host defense against microbial pathogens [69]. In MPV, D6L encodes the host homolog of interleukin-18 (IL-18) binding protein, which plays a major role in evading the host immune response [70]. 2006, Meng et al. identified L5, K53 and S55 of hIL-18 as key sites for IL-18 binding through mutagenesis experiments, and ensured that the virus evaded immune recognition by competitively occupying the binding site [71]. In 2008, Brian et al. proposed three potential binding sites based on the crystal structure and analyzed the hydrophobic and hydrogen bonds on the cavity surface, which provided a theoretical basis for the rational design of inhibitors against orthopoxvirus [72, 73].

**3.2.8 E13L.** The assembly of viral particles is an extremely complex process involving dozens of proteins including E13L, C16L, A11L, and M2R. E13L, a trimeric capsid protein, is the first contact protein for viral assembly and can stabilize on lipid membranes to enhance membrane rigidity [44, 74 and 75]. In 2007, Charity et al. found that rifampicin could achieve inhibition by binding to E13L, but the inhibition was reversible and the virus resumed assembly after a few minutes of discontinuation [76]. Despite the shortcomings of rifampicin in the treatment of poxvirus infection, its binding to E13L provides a potential active docking target for subsequent drug design based on viral assembly [77].

**3.2.9 A22R.** The identification of molecules capable of interacting with the replication complex and inhibiting its activity is a promising strategy for the design of new antiorthopox viral drugs. The first to be mentioned in the DNA replication process of MPV is A22R. First, it can form a complex with D4 to promote the formation of heterodimeric processing factors of viral DNA [59]. Secondly, A22R can form a bridge between D4 and DNA polymerase by interacting directly with F8L [78]. In addition, A22R can also interact with E5R and H5R, thus participating in the transcription and elongation of the protein [79, 80]. As a core component, A22R is a key component in the formation/ stabilization of DNA complexes, the discovery of molecules able to inhibit the function of A22R could represent a relevant strategy to generate new compounds against orthopoxviruses.

**3.2.10 I7L.** In 2002, the gene products of I7L ORF have been shown to be cysteine proteases that cleave the core protein precursors and membrane proteins of the virus [81]. As ideal therapeutic targets, drugs designed against I7L can effectively block the cleavage of core protein precursors and the subsequent maturation of immature viral intermediates into infectious

intracellular mature viruses [82, 83]. Two years later, Byrd et al. screened a library of compounds containing 51,000 inhibitors and found that ttp-6171 was effective in inhibiting the biological activity of I7L without significant cytotoxicity or inhibition of the growth of other organisms [41].

**3.2.11 C19L.** The formation of EEVs involves many viral genes and presents opportunities for the development of specific inhibitors. C13L is a palmitoylated membrane protein that binds to post-Golgi vesicles and then recruits B6R and A38R together to promote the EV envelope process; in addition, it also acts as a mediator to promote fusion during viral invasion of cells [44, 84]. More recently, with the spread of MPV, C13L was reintroduced and a specifically recognized inhibitor, tecovirimat (ST-246) was found to be effective in inhibiting its activity, and became the only approved drug used for poxvirus treatment. ST-246 can effectively inhibit orthopoxvirus (VAR, VAC, CPV and MPV) and also prevents various mutations. More importantly, it is highly safe and tolerable without serious side effects [46, 85].

**3.2.12 A50R.** In the design of drugs for monkeypox, targeting the process of structural protein processing and mature progeny virions assembly to develop drugs is another important pathway [39, 86]. A50R: DNA ligase, which is deeply involved in the process of virus processing and assembly, is considered as a valuable research target [46]. In 2007, Deng et al. identified 13 compounds that can effectively inhibit poxvirus replication by screening the drug library, and finally identified mitoxantrone, which has no significant effect on viral gene expression, as the optimal inhibitor [87]. Nevertheless, the specific inhibition mechanism needs to be further investigated.

## 3.3 Construction of inhibitor library

In recent years, scientists around the world have conducted a variety of studies to develop and design novel monkeypox inhibitors [46, 85, 88–99]. In summary, there are five main drug discovery approaches [46, 93, 99 and 100, 101]: first, to design new drugs from existing drugs; second, to discover new monkeypox inhibitors from known drug molecule structures; third, to explore potential drug molecules based on other orthopoxviruses; fourth, to find novel drug structures through computer-aided drug design (CADD) methods; and fifth, to use old drugs to obtain possible drugs through virtual screening. In response to the new round of MPV epidemic, it is urgent to find drugs that can effectively combat the epidemic, and the search for drugs through orthopoxvirus drugs can ensure the toxic side effects of drugs while greatly guaranteeing the safety, and then combine with CADD to reduce the drug discovery time. Thus, the combination of multiple methods can provide the most effective method for the discovery of monkeypox drugs.

In our work, we reviewed articles from the last 50 years and screened the existing databases (ChEMBL, DrugBank, PubChem, BindingDB) to construct a database containing 189 potential drug molecules (S2 Table). By backtracking, we also summarized the inhibitors' viruses of action, target genes/ proteins, possible mechanisms of action, inhibitory activity and cited literature, and provided their structures to provide a basis for subsequent relevant studies. Since the 1970s, studies on orthopox viruses have been emerging, Yoshii and Tagaya have discovered the activity of 5-bromodeoxyuridine and Cytosine arabinoside against the virus [1, 9]. Among them, 5-bromodeoxyuridine acts mainly on MPV and VAC and achieves inhibition by affecting virus replication, while Cytosine arabinoside has three possible mechanisms of inhibition: (a) blocking the reduction of cytidine diphosphate to deoxycytidine diphosphate, (b) incorporation into DNA, thereby producing faulty DNA, and (c) inhibition by competition with deoxycytidine triphosphate on DNA polymerase water. Towards the beginning of the 21st century, orthopoxvirus research entered a period of rapid development, and a drug with strong activity

against orthopoxvirus genera including VAC, CPV, VAR and MPV, tecoviride (ST-246), was discovered by yang et al [2]. ST-246 effectively blocks intracellular viruses in the endoplasmic reticulum tegument, preventing the virus from being released from the cell and reducing the chance of infecting other normal cells, thus achieving a therapeutic. ST-246 effectively blocks intracellular virus in the endoplasmic reticulum tegument, preventing the virus from being released from the cell and reducing the chance of infecting other normal cells, thus achieving treatment [102]. The therapeutic effect is exerted in two ways: stopping the virus from multiplying inside the cell and reducing the lethality of smallpox; and preventing the virus from shedding from carriers and reducing the transmission rate [103]. In recent years, the rise of MPV has led to a flurry of research on orthopox viruses: Wu et al. exploiting two VAC strains containing firefly luciferase (VTT-Fluc and VG9-Fluc) as surrogate viruses performed a high-throughput screening of 767 approved drugs and determined that Mycophenolate mofetil (MMF) and tranilast (TRA) are promising anti-smallpox virus candidates for further optimization and repurposing for use in clinical practice [3]; Sokolova et al. designed and synthesized a series of the bornyl ester/ amide derivatives with N-containing heterocycles and identified six with higher bioactivity with VAV by determining the relationship between VAC Six compounds with superior inhibitory activity and less toxic side effects were identified by measuring the biological activity with VAV [11].

## 3.4 Homology modeling and preliminary verification of poxvirus proteins

The three-dimensional structure of proteins is an important basis for understanding their biological functions and for structure-based drug design [104]. With the rapid development of structural biology, the speed of measuring protein 3D structures using methods such as NMR, X-Ray or cryoelectron microscopy (Cryo-EM) crystallography has been greatly improved [39, 105]. However, the 3D structures of some proteins are still difficult to determine due to many reasons, such as excessive molecular weight or difficulty in crystallization. Before modeling the protein structures, we first performed a series of physicochemical property analyses on 12 potential target sequences, including Weight, PI, Extinction, Instability, Aliphatic and Hydrophathicity, and compared the sequences of monkeypox with those of human origin and other orthopoxviruses (S1 Table). It was found that the structural domains of the proteins were stabilized to a certain extent and could be used for subsequent studies; The sequence similarity of all proteins except D6L was less than 60% to human, which avoided the side effects of the drug caused by host homology. D6L, as an interleukin-18 binding protein homologous to the host, functionally affirmed its high sequence similarity to human proteins, a characteristic that also increased the difficulty of developing drugs against it; finally, the sequence similarity between monkeypox virus and orthopoxvirus was higher than 78%, proving the existence of commonality between them, and providing an opportunity to seek possible orthopoxvirus-based drugs for the subsequent studies. This lays a theoretical foundation for the search of possible monkeypox virus drugs based on orthopoxviruses.

For the key proteins of monkeypox, we found that eight (A49R, C3L, H1L, E4R, C7L, A41L, D6L and E13L) could be obtained from orthopox viruses (VAR, VAC and MPV) modeled with high sequence similarity. As for the key proteins (A22R, I7L, C19L and A50R), which are not present in any of the orthopoxvirus genera, we chose AlphaFold2, which is only one atomic width away from the true structure for most of the predicted protein structures, to predict them. Notably that based on the multifaceted superiority, AlphaFold2 reaches the level of prediction observed by sophisticated instruments such as cryoelectron microscopy, which can effectively guarantee the accuracy of the predicted structures (Fig 2) [106]. Hence, we constructed a database of orthopoxvirus proteins with a capacity of 45.

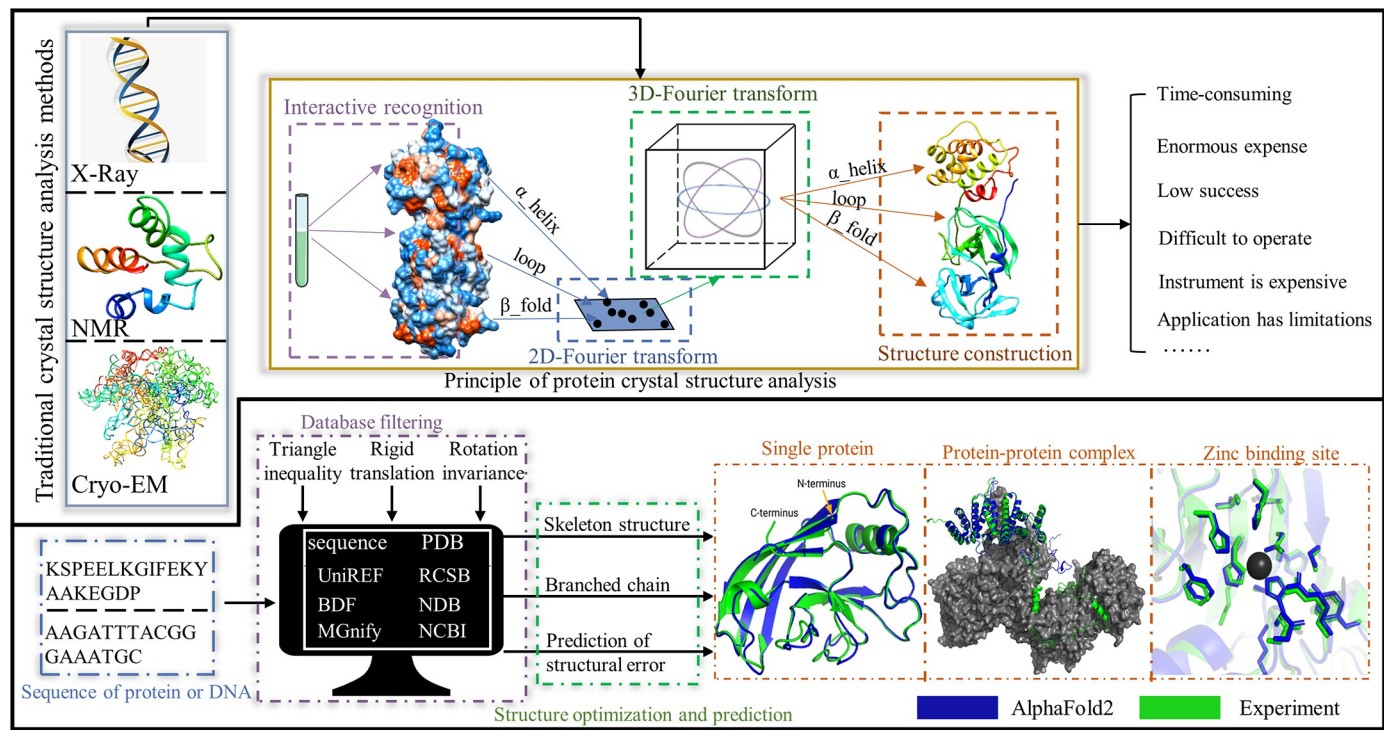

**Fig 2. Comparison of two crystal structure prediction methods: Traditional crystal structure acquisition methods (top) and AI-based crystal structure prediction (bottom).**

Table 2 gives the methods for obtaining the 12 key protein structures, the types of modeled protein structures and the scoring results of the AplhaFold2 modeled structures. It can be found that basically all the orthopoxvirus genus proteins can be reasonably molded and the simulated structures are relatively reasonable. S1–S16 Figs show the reasonableness of the modeled structures of each system. Taking the A22R modeling results as an example (S9 and S10 Figs), in the AlphaFold2 modeling results, rank_4 has the highest pLDDT and Ptmscore for the five modeled structures given, which means that this structure is the most representative of the dominant conformation of A22R. Specifically, from the Sequence coverage and predicted IDDT, it is easy to find that the residues of the protein are in high Sequence identity

**Table 2. Physicochemical properties of 12 potential target protein sequences and sequence similarity to human and orthopoxvirus.**

|  | A49R | H1L | C7L | D6L | A22R | C19L | C3L | E4R | A41L | E13L | I7L | A50R |
|---|---|---|---|---|---|---|---|---|---|---|---|---|
| Weight | 23289.7 | 19727.8 | 25705.3 | 14337.6 | 49147.1 | 41808.7 | 4982.7 | 25076.8 | 25371.9 | 61937.0 | 79605.2 | 63574.3 |
| PI | 5.3 | 9.2 | 4.7 | 6.5 | 5.6 | 6.5 | 4.3 | 7.0 | 5.0 | 5.2 | 7.5 | 7.5 |
| Extinction | 28420 | 17880 | 28310 | 18450 | 49280 | 52370 | 8940 | 45380 | 24870 | 58680 | 72660 | 58790 |
| Instability | 50.4 | 46.8 | 41.4 | 29.0 | 36.3 | 29.8 | 29.7 | 42.1 | 39.4 | 28.8 | 30.0 | 32.2 |
| Aliphatic | 90.8 | 86.0 | 84.1 | 82.7 | 92.8 | 93.8 | 95.3 | 92.1 | 81.9 | 90.1 | 86.4 | 90.0 |
| Hydrophathicity | -0.245 | -0.308 | -0.248 | -0.247 | -0.258 | -0.112 | 0.160 | -0.258 | -0.360 | -0.200 | -0.236 | -0.344 |
| Human Identity | 42.8 | 26.9 | <10% | 94.3 | <10% | 16.6 | 24.2 | <10% | <10% | 26.7 | 19.6 | 56.6 |
| VAR Identity | 98.5 | 98.8 | 83.6 | 92.1 | 96.7 | 97.3 | 78.6 | 96.8 | 91.4 | 98.9 | 98.8 | 96.4 |
| VAC Identity | 99.0 | 100.0 | 83.6 |  | 97.4 | 98.7 | 95.2 | 98.6 | 94.1 | 98.9 | 99.1 | 97.8 |
| CPV Identity | 99.0 | 98.8 | 87.2 | 79.7 | \ | 98.7 | 97.6 | \ | 94.1 | \ | \ | 97.8 |

**Table 3. Structure acquisition methods, structural soundness scoring and docking success rate of 12 key proteins.**

| ORFs | Methods | Protein Category | pLDDT | Rate%* | ORFs | Methods | Protein Category | pLDDT | Rate% |
|------|---------|------------------|-------|--------|------|---------|------------------|-------|-------|
| A49R | RCSB, DS | **VAR**, VAC, CPV, MPV | - | 78.79 | C3L | RCSB, DS | VAR, **VAC**, CPV | - | 82.32 |
| H1L | RCSB, DS | VAR, **VAC**, CPV, MPV | - | 68.19 | E4R | RCSB, DS | VAR, **VAC**, CPV, MPV | - | 77.78 |
| C7L | RCSB, DS | **VAR**, VAC, CPV, MPV | - | 83.33 | A41L | RCSB, DS | **VAR**, VAC, CPV, MPV | - | 73.23 |
| D6L | RCSB, DS | VAR, VAC, CPV, MPV | - | 79.30 | E13L | RCSB, DS | VAR, **VAC**, CPV, MPV | - | 76.26 |
| A22R | AlphaFold2, DS | VAR, VAC, **MPV** | 90.3 | 85.29 | I7L | AlphaFold2, DS | VAR, VAC, CPV, **MPV** | 86.9 | 82.32 |
| C19L | AlphaFold2, DS | VAR, VAC, **MPV** | 91.6 | 75.25 | A50R | AlphaFold2, DS | VAR, VAC, CPV, **MPV** | 91.2 | 69.70 |

*: Proportion of docking results with docking affinity below -0.5 in the small molecule database

and more than 98 residues are in high pIDDT values (pIDDT > 60) [107]. Based on rank_4, we used Discovery Studio to model the VAR, VAC, and MPV, and the results showed that over 95% of the amino acids were in the conformational optimum region, and only a few amino acids (VAR:5, VAC:7, MPV:6) were in the conformational inappropriate region. Considering that the protein contains more than 400 amino acids and the amino acids in the unreasonable region are in the loop region, which is far from the core region of the protein, it is considered reasonable to build the conformation [28, 108]. In addition, further reverse molecular docking work was used to verify the rationality of the protein structure. Through the Autodock vina program package, we docked 189 small molecules to each protein separately. As shown in Table 3, more than 135 (more than 68%) were docked with affinity less than -5 for each system, indicating that the protein active site is relatively reasonable and can ensure the follow-up work.

## 3.5 Webtool of MPV

Monkeypox virus Docking Server is a platform for screening potential key proteins and possible inhibitors of monkeypox, enabling rapid molecular docking or reverses molecular docking between proteins and small molecules, accelerating the search for novel inhibitors and driving the identification of key protein targets. Under our platform, there are two main components: screening of potential inhibitors and protein target validation. In the inhibitor screening part, the molecular docking approach is used, with two modes of single and multiple molecule docking. Users can use the prepared key protein database to dock the provided small molecules with the specified proteins rapidly (the number of docking is 1), and screen the potential compounds with better binding based on the docking results; then set the number of docking to 100 through fine screening to further determine the binding mode of the substrate and provide theoretical guidance for subsequent experiments. In the protein target identification part, using reverse molecular docking, two main tasks can be achieved: for proteins that do not have target information, potential active sites can be obtained; for systems with clear active centers, potential precursor drugs can be screened by docking. Through the provision of a large number of orthopoxvirus genus inhibitors, the platform performs non-stop docking for user-supplied proteins and provides the 10 best results according to scoring. In addition, the docking results can be visualized in 3D by JSmol, and the user can view and download the docking results from the result page. Up to now, the average number of visitors to the website for 7 days is over 200, and the countries visited are mainly China, Germany and the United States.

Here, we take A49R and the inhibitor database as an example for screen potential inhibitors. Select "Ligand" in "Submit" and select A49R receptor in protein target (select monkeypox virus type). Upload the small molecule files (mol2, smi, sdf) and choose the

| Rank | 1 | 2 | 3 | 4 | 5 | 6 | 7 | 8 | 9 | 10 |
|------|------|------|------|------|------|------|------|------|------|------|
| **Score** | -8.00 | -7.60 | -7.40 | -7.00 | -6.90 | -6.80 | -6.80 | -6.80 | -6.60 | -6.60 |

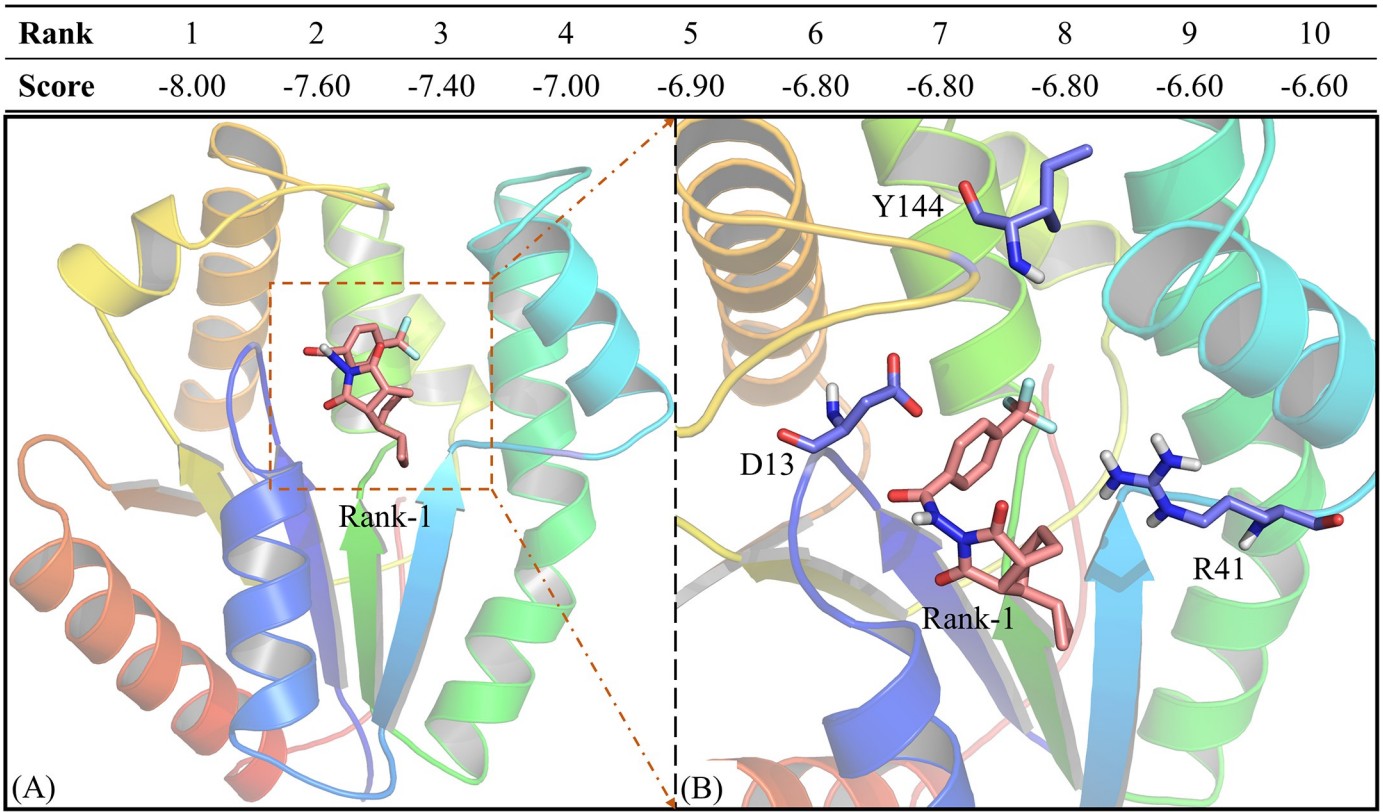

**Fig 3. The summary of top 10 docking models and the three-dimensional structure of rank-1_protein complex.**

exhaustiveness configuration. In addition, complete the user's email in order to receive reminders after task completion. Fig 3 presents the docking results and it can be found that the first 10 small molecules are able to dock properly with the protein (Score < -6.60). In the previous study, we know that the active center of A49R is composed of three residues, D13, R41 and Y144, thus we further show the positions of these three residues (Fig 3B). It is not difficult to find that the ligand is present in the active pocket and is adjacent to the three key residues, which also proves the accuracy of platform docking. For protein target identification or potential drug screening systems, we provide a selection of proteins in the "Submit" section. In addition to selecting the Exhaustiveness configuration and filling in the email address, the user needs to upload the protein file, import the docking center coordinates and docking pocket size. The result is the same as in "Ligand", where users can find the optimal small molecule based on the docking result and predict the potential target location according to the docking site.

## 4. Discussion

The MPV Docking Server, an online meta-server for key protein target prediction and potential drug screening for MPV, was built using Autodock Vina as the docking engine, targeting MPV as well as other orthopox viruses (VAR, VAC and CPV). Compared to other docking servers of the same type, this server provides a user-friendly interface and binding mode visualization for the results, rendering it a useful tool for orthopox virus drug and key protein discovery. In addition, through extensive literature and database screening, three virtual

databases were constructed: MPV open reading frame, orthopox virus key protein library and inhibitor database, which provide theoretical guidance for subsequent development based on MPV and orthopox virus and greatly accelerate the discovery of therapeutic drugs for monkeypox and address the current critical situation.

Of course, there are still some shortcomings in the current website, including limited capacity of key protein libraries and small molecule inhibitors, the drug screening method still has scope for improvement, and the lack of experimental validation. In the subsequent maintenance of the website, we will still improve the system, the improvement measures mainly include: 1 expand the sample capacity, dig more key proteins of MPV and continue to collect potential inhibitors; 2 optimize the docking method, fully consider the solvent effect, spatial site resistance, van der Waals interactions and other factors in the screening process; 3 cooperation and assistance, cooperate with excellent laboratories at home and abroad to further improve the experimental system: based on luciferase screening method, using an eight-point dose response format assay to detect antiviral activity; real-time reverse transcriptase-PCR and western blotting based on the specific expression of key proteins; antiviral activity against MPV was determined by RNA interference assay by reduction of empty spots and virus yield. In this way, we aim to obtain novel drugs that are effective against MPV and to promote the development of monkeypox treatment by expanding the sample and optimizing the screening method with the aid of experiments.

## Supporting information

**S1 Table. Physicochemical properties of 12 potential target protein sequences and sequence similarity to human and orthopoxvirus.**
(DOCX)

**S2 Table.**
(XLSX)

**S1 Fig. Ramachandran plots of A49R model.**
(TIF)

**S2 Fig. Ramachandran plots of C3L model.**
(TIF)

**S3 Fig. Ramachandran plots of H1L model.**
(TIF)

**S4 Fig. Ramachandran plots of E4R model.**
(TIF)

**S5 Fig. Ramachandran plots of C7L model.**
(TIF)

**S6 Fig. Ramachandran plots of A41L model.**
(TIF)

**S7 Fig. Ramachandran plots of D6L model.**
(TIF)

**S8 Fig. Ramachandran plots of E13L model.**
(TIF)

**S9 Fig. Rationalisation of 5 structures (A22R) based on AlphaFold2 modeling as well as the scoring of pLDDT and Ptmscore.**
(TIF)

**S10 Fig. Ramachandran plots of A22R model.**
(TIF)

**S11 Fig. Rationalisation of 5 structures (I7L) based on AlphaFold2 modeling as well as the scoring of pLDDT and Ptmscore.**
(TIF)

**S12 Fig. Ramachandran plots of I7L model.**
(TIF)

**S13 Fig. Rationalisation of 5 structures (C19L) based on AlphaFold2 modeling as well as the scoring of pLDDT and Ptmscore.**
(TIF)

**S14 Fig. Ramachandran plots of C19L model.**
(TIF)

**S15 Fig. Rationalisation of 5 structures (A50R) based on AlphaFold2 modeling as well as the scoring of pLDDT and Ptmscore.**
(TIF)

**S16 Fig. Ramachandran plots of A50R model.**
(TIF)

## Acknowledgments

The authors are grateful to Dr. Liu Wei for her help with the preparation of figures in this paper.

## Author Contributions

**Conceptualization:** Huaichuan Duan, Quanshan Shi, Zelan Zhang.

**Data curation:** Huaichuan Duan.

**Formal analysis:** Yujie Cao.

**Funding acquisition:** Jianping Hu.

**Methodology:** Xinru Yue.

**Project administration:** Li Liang.

**Resources:** Ling Liu.

**Software:** Quanshan Shi, Yueteng Wang.

**Validation:** Zuoxin Ou.

**Writing – original draft:** Huaichuan Duan.

**Writing – review & editing:** Jianping Hu, Hubing Shi.

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
