## [Decision Letter · Decision Letter 0]

8 Mar 2024

PONE-D-24-03917Development of mechanism‐based inhibitor against Monkeypox virus: A WEB to predict the binding patternsPLOS ONE

Dear Dr. Hu,

Thank you for submitting your manuscript to PLOS ONE. After careful consideration, we feel that it has merit but does not fully meet PLOS ONE’s publication criteria as it currently stands. Therefore, we invite you to submit a revised version of the manuscript that addresses the points raised during the review process.

We look forward to receiving your revised manuscript.

Kind regards,

Mahmoud Kandeel

Academic Editor

PLOS ONE

“The National Key Research and Development Program of China (No. 2022YFD2200101-04).”

5. We note that Figure s17 in your submission contain [map/satellite] images which may be copyrighted. All PLOS content is published under the Creative Commons Attribution License (CC BY 4.0), which means that the manuscript, images, and Supporting Information files will be freely available online, and any third party is permitted to access, download, copy, distribute, and use these materials in any way, even commercially, with proper attribution. For these reasons, we cannot publish previously copyrighted maps or satellite images created using proprietary data, such as Google software (Google Maps, Street View, and Earth). For more information, see our copyright guidelines: http://journals.plos.org/plosone/s/licenses-and-copyright.

a. You may seek permission from the original copyright holder of Figure s17 to publish the content specifically under the CC BY 4.0 license. 

6. We notice that your supplementary figures are included in the manuscript file. Please remove them and upload them with the file type 'Supporting Information'. Please ensure that each Supporting Information file has a legend listed in the manuscript after the references list.

Reviewers' comments:

Reviewer's Responses to Questions

**Comments to the Author**

1. Is the manuscript technically sound, and do the data support the conclusions?

Reviewer #1: Partly

Reviewer #2: Partly

2. Has the statistical analysis been performed appropriately and rigorously? 

Reviewer #1: No

Reviewer #2: No

3. Have the authors made all data underlying the findings in their manuscript fully available?

Reviewer #1: No

Reviewer #2: No

4. Is the manuscript presented in an intelligible fashion and written in standard English?

Reviewer #1: Yes

Reviewer #2: Yes

5. Review Comments to the Author

Reviewer #1: In this study the authors claim to have developed a monkeypox core protein library and a library of inhibitors targeting these core proteins. The authors have in addition built a docking server that predicts the binding modes between the protein and the substrate to understand the molecular dialogues between these entities. Currently there are hardly such servers dedicated for monkeypox virus. Therefore, such an endeavour is greatly welcome. Although the paper appears to be sound overall, I have some key recommendations that that I think can be incorporated before the paper can be accepted for publication.

1. In the methodology section method 2.1.Open reading frame (ORF), the ORF needs to be defined more concretely. I advise the authors to have a look at the results section (3.1 The meaning of the open reading frame (ORF) and core proteins in this study) of the paper titled “Identification of core therapeutic targets for Monkeypox virus andrepurposing potential of drugs against them: An in silico approach”for better clarity.

2. Similarly, the term core protein creates a lot of confusion. From the view of a genomics researcher core protein is more closely associated with proteins that are highly conserved across all the genomes. However, from the point of view of a classical virologist core protein usually means the structural protein embedded on the outer membranes that give a virus its definitive shape. Therefore, the authors should clearly mention this in the manuscript as readers will comprise of people from multiple disciplines. Again, I advise the authors to have a look at the results section (3.1 The meaning of the open reading frame (ORF) and core proteins in this study) of the paper titled “Identification of core therapeutic targets for Monkeypox virus and repurposing potential of drugs against them: An in silico approach”for better clarity.

3. Shared host homology is one of the biggest reasons for the failure of a drug from reaching the bed from the bench. This often creates adverse reactions by to off-targeting the host proteins instead. As a consequence, drugs often fail during clinical trials. Therefore, the authors must verify that the 12 key proteins do not share homology with human proteins beyond an allowable threshold (usually e-value of 0.0001 against the entire set of human proteome dataset (taxid: 9606) is considered a good threshold). The authors must provide this information in the manuscript.

4. Why only 12 proteins were selected as targets? Are all the 12 key proteins druggable? The authors need to verify this information as not all proteins present highly druggable surfaces for docking. The validity of the structures used for docking should be more adequately demonstrated.

5. Are any of the proteins intrinsically disordered? This is a very key question that needs answering. They are a fascinating class of proteins that lack a fixed or well-defined three-dimensional structure in their free state. Therefore, docking against such proteins are more challenging than what meets the eye. Therefore, physiochemical properties like isoelectric point (pI), molecular weight, aliphatic index, instability index, extinction co-efficient and residues accessibility of the 12 key proteins must be examined and provided in the manuscript. This data will aid experimental biologists to choose suitable proteins for experimental validation.

Overall, the proteins that are highly conserved, non-host homologous, highly druggable are usually considered very high value therapeutic targets. The proteins that ensure these features are likely to have a higher rate of success. Hence, I feel these key points need to be investigated by the authors before being considered for final consideration. The above recommended paper should aid the authors in this endeavour.

6. The efficiency of the docking server is not demonstrated with appropriate controls. Since, none of the results are experimentally validated, it is necessary to demonstrate the efficiency by comparison with other docking models.

Reviewer #2: The paper titled “Development of mechanism‐based inhibitor against Monkeypox virus: A WEB to predict the binding patterns” is like a review on the said “Monkeypox virus Docking Server”. The server might serve as a good resource for people working on the virus, which is indeed comprehensive and well-thought-out. It can perform modeling, docking, and annotation. Still, it lacks a firm basis, and improvement is needed. The issues pointed out below are critical and need to be properly addressed:

1. The key to publishing a paper on a developed tool is to put forth a detailed analysis concerning the accuracy of the results yielded by it through statistical inferences. The authors have presented an example but it's not enough to validate the potentiality of the server.

2. It is clear that the server provides functional annotations for specific genes or proteins of interest, but are there tools or functionalities available for analyzing gene function, protein structure, or pathway enrichment? How does the server facilitate the comparative analysis of genes, proteins, or genomes across different ortho-poxviruses? It would be interesting if there were such a provision that the server could carry out.

3. Have the authors conducted site-specific docking or blind docking methodologies in this investigation? Please provide a clear explanation. It's not clear how, in case of no active site details, the tool can locate an active site for a newly modeled protein by reverse docking. The authors need to elaborate on that.

4. Given that a significant portion of the protein crystal structures remains unresolved, the authors generated homology models to address this limitation. How did the authors predict the orthosteric sites of the protein targets? Was a validation conducted against known inhibitors to assess the accuracy of the predicted orthosteric sites?

5. If the authors have not undertaken validation utilizing established inhibitors against the protein targets, I recommend conducting a benchmark study for comprehensive evaluation.

6. What rationale led the authors to choose AutoDock Vina? Kindly elucidate the justification for this selection. Additionally, is it feasible to conduct reverse screening of peptide-like inhibitors using the web server?

7. The authors must mention how their server is unique/better than similar tools developed in the past, such as the 'COVID-19 Docking Server'.

8. In the literature study in the introduction, the authors should also cite one recent exciting work published in ACS JPCL, where the authors modeled the P37 protein of MPXV using the AF tool and performed computer-aided drug discovery. The paper suits the context of the webserver as well the manuscript. https://doi.org/10.1021/acs.jpclett.3c00087

9. Additionally, the authors are advised to improve the website to make it more user-friendly:

a. The user interface must be made to look interesting and organized. Various visualization tools are now available to present 3D protein structures and interactions. The authors should work on it.

b. The authors should think of a simple one/two-worded name for the server, implying that it is a docking server meant only for MPXV proteins and also serves as a comprehensive database.

c. The manuscript title can also be improved. It does not resonate with the work done or the core feature of the server.

Evaluating the quality of the work as well as the existing drawbacks, the paper can be recommended for publication in PLOS ONE only after the above-mentioned revisions. Apparently, this is the first MPXV webserver. The authors should do major work to improve it and publish it.

6. PLOS authors have the option to publish the peer review history of their article (what does this mean?). If published, this will include your full peer review and any attached files.

Reviewer #1: **Yes: **Bharat Bhusan Subudhi

Reviewer #2: No

---

## [Author Response · Author response to Decision Letter 0]

1 Apr 2024

Prof. Jianping Hu, PhD

Chengdu University 

610106 Chengdu, China

E-mail: hjpcdu@163.com

Chengdu, January 18th, 2024

Dear Editorial Office

Editor, PLOS ONE

Thank you very much for your email of Major decision, regarding our manuscript Development of mechanism‐based inhibitor against Monkeypox virus: A WEB to predict the binding patterns (Manuscript Number: PONE-D-24-03917R1).

We would like to thank wholeheartedly the referees for their valuable and helpful comments about our manuscript. Thanks to the reviewers for their recognition of the research objectives, rationale and methods of the article, as well as confirming the experimental data of the article. Suggestions on the figures and structure of the article will also be listened to and improved upon. Our replies to reviewers’ comments are enclosed below and in the revised version of the manuscript, we marked two parts in red: the suggestions given by the reviewers and the contents modified by the second reading of the article. We hope that our revised manuscript is now suitable for publication in PLOS ONE.

Yours sincerely,

Dr. Jianping Hu

E-mail: hjpcdu@163.com

Reply to Editor

Thank you to the editors for their advice on article acceptance. We have reconfirmed PLOS-ONE's requirements for articles as well as shared code to ensure that the submitted manuscripts are in order. Second, the funder had no role in the study design, data collection and analysis, publication decision, or preparation of the manuscript. Additionally, the editor mentioned a data sharing program, and we accept the editor's suggestion that we can accept sharing of data prior to article acceptance. The editor also raised the issue of copyright, and we were acutely aware of the importance of copyright, so we corrected the article and removed Figure s.17 Finally, for the supporting material we made a separate file and made sure that each information file had a legend in the manuscript following the list of references.

Reply to Reviewer 1

In this study the authors claim to have developed a monkeypox core protein library and a library of inhibitors targeting these core proteins. The authors have in addition built a docking server that predicts the binding modes between the protein and the substrate to understand the molecular dialogues between these entities. Currently there are hardly such servers dedicated for monkeypox virus. Therefore, such an endeavor is greatly welcome. Although the paper appears to be sound overall, I have some key recommendations that that I think can be incorporated before the paper can be accepted for publication.

Specific comment 1

In the methodology section method 2.1. Open reading frame (ORF), the ORF needs to be defined more concretely. I advise the authors to have a look at the results section (3.1 The meaning of the open reading frame (ORF) and core proteins in this study) of the paper titled “ Identification of core therapeutic targets for Monkeypox virus and repurposing potential of drugs against them: An in silico approach ”for better clarity.

Response 1

I appreciate the editor's comments and I have taken the time to check out the full article. The Open Reading Frame (ORF) serves as the beginning of the article and has an important role in the subsequent work. In the Methods section we considered the length of the article, so we simply explained the ORF. In response to the editor's suggestion, we have explained this section in more detail, and the specific corrections are as follows:

ORF is a fundamental concept in molecular bioinformatics, it is the normal nucleotide sequence of a structural gene, the reading frame from the start codon to the stop codon can encode a complete polypeptide chain, between which there is no stop codon that interrupts the translation. The term open refers to the "open" region of an intact gene used for protein translation, and reading frame refers to one of the six possibilities for translation of a double-stranded gene sequence into amino acids. In molecular biology, the detection of ORFs is an important step in the discovery of specific protein-coding genes in the genome sequence, and can serve as an indicator of a potential protein-coding gene. In our work, the acquisition of ORF mainly includes literature reading and database research.

In addition to the suggestions for ORFs, the editors also made a suggestion to change the title of the article to "Identification of core therapeutic targets for Monkeypox virus and repurposing potential of drugs against They: An in-silico approach", which is more in accordance with the content of the article. In our very first title, we targeted the server, ignoring the fact that the article studies more on monkeypox virus and core proteins. Seeing the editor's suggestion, we appreciate being able to identify this issue and also think that changing the title would benefit the article more.

Specific comment 2

Similarly, the term core protein creates a lot of confusion. From the view of a genomics researcher core protein is more closely associated with proteins that are highly conserved across all the genomes. However, from the point of view of a classical virologist core protein usually means the structural protein embedded on the outer membranes that give a virus its definitive shape. Therefore, the authors should clearly mention this in the manuscript as readers will comprise of people from multiple disciplines. Again, I advise the authors to have a look at the results section (3.1 The meaning of the open reading frame (ORF) and core proteins in this study) of the paper titled “Identification of core therapeutic targets for Monkeypox virus and repurposing potential of drugs against them: An in-silico approach” for better clarity.

Response 2

We thank the reviewers for this suggestion, and we do recognize the reviewers' suggestion that the core proteins were improperly worded. In our work, core proteins are used to denote potential proteins with the potential to inhibit monkeypox virus, and do not refer to structural proteins or highly conserved proteins as mentioned by the reviewer. In order to avoid misunderstanding, we have corrected all references to core proteins in the article and replaced the original core proteins with potential targets. The following are some of the corrections:

A new round of monkeypox virus has emerged in the United Kingdom since July 2022 and rapidly swept the world. Currently, despite numerous research groups are studying this virus and seeking effective treatments, the information on the open reading frame, inhibitors, and potential targets of monkeypox has not been updated in time, and the comprehension of monkeypox target ligand interactions remains a key challenge. Here, we first summarized and improved the open reading frame information of monkeypox, constructed the monkeypox inhibitor library and potential targets library by database research as well as literature search, combined with advanced protein modeling technologies (Sequence-based and AI algorithms-based homology modeling). In addition, we build monkeypox virus Docking Server, a web server to predict the binding mode between targets and substrate. The open reading frame information, monkeypox inhibitor library, and monkeypox potential targets library are used as the initial files for server docking, providing free interactive tools for predicting ligand interactions of monkeypox targets, potential drug screening, and potential targets search. In addition, the update of the three databases can also effectively promote the study of monkeypox drug inhibition mechanism and provide theoretical guidance for the development of drugs for monkeypox.

In this issue, the reviewer again proposed a change in the title of the article. In the previous comment, we clearly stated that we recognized the reviewer's title and explained the reason for the title in the first place. Again, we thank the reviewers for their attention to this issue.

Specific comment 3

Shared host homology is one of the biggest reasons for the failure of a drug from reaching the bed from the bench. This often creates adverse reactions by to off-targeting the host proteins instead. As a consequence, drugs often fail during clinical trials. Therefore, the authors must verify that the 12 key proteins do not share homology with human proteins beyond an allowable threshold (usually e-value of 0.0001 against the entire set of human proteome dataset (taxid: 9606) is considered a good threshold). The authors must provide this information in the manuscript.

Response 3

This question raised by the reviewer is very interesting and we have addressed it with a detailed explanation. Shared host homology as an important issue that must be considered for subsequent drug development, if the homology is too high it will lead to poor drug selectivity, and attacking the viral cells while also harming the host cells to a certain extent. In our article, this issue was neglected, so in the revised manuscript, for both host and viral cells we added this part. This includes calculation of physicochemical information of 12 proteins, sequence similarity with human-derived proteins, VAC, VAR and CPV, detailing the host homology issue.

The three-dimensional structure of proteins is an important basis for understanding their biological functions and for structure-based drug design [104]. With the rapid development of structural biology, the speed of measuring protein 3D structures using methods such as NMR, X-Ray or cryoelectronic microscopy (Cryo-EM) crystallography has been greatly improved [39, 105]. However, the 3D structures of some proteins are still difficult to determine due to many reasons, such as excessive molecular weight or difficulty in crystallization. Before modeling the protein structures, we first performed a series of physicochemical property analyses on 12 potential target sequences, including Weight, PI, Extinction, Instability, Aliphatic and Hydrophathicity, and compared the sequences of monkeypox with those of human origin and other orthopox viruses (table s1). It was found that the structural domains of the proteins were stabilized to a certain extent and could be used for subsequent studies; The sequence similarity of all proteins except D6L was less than 60% to human, which avoided the side effects of the drug caused by host homology. D6L, as an interleukin-18 binding protein homologous to the host, functionally affirmed its high sequence similarity to human proteins, a characteristic that also increased the difficulty of developing drugs against it; finally, the sequence similarity between monkeypox virus and orthopoxvirus was higher than 78%, proving the existence of commonality between them, and providing an opportunity to seek possible orthopoxvirus-based drugs for the subsequent studies. This lays a theoretical foundation for the search of possible monkeypox virus drugs based on orthopox viruses.

For the key proteins of monkeypox, we found that eight (A49R, C3L, H1L, E4R, C7L, A41L, D6L and E13L) could be obtained from orthopox viruses (VAR, VAC and MPV) modeled with high sequence similarity. As for the key proteins (A22R, I7L, C19L and A50R), which are not present in any of the orthopoxvirus genera, we chose AlphaFold2, which is only one atomic width away from the true structure for most of the predicted protein structures, to predict them. Notably that based on the multifaceted superiority, AlphaFold2 reaches the level of prediction observed by sophisticated instruments such as cryoelectron microscopy, which can effectively guarantee the accuracy of the predicted structures (Fig.2) [106]. Hence, we constructed a database of orthopoxvirus proteins with a capacity of 45.

Table s1 Physicochemical properties of 12 potential target protein sequences and sequence similarity to human and orthopoxvirus

 A49R H1L C7L D6L A22R C19L C3L E4R A41L E13L I7L A50R

Weight 23289.7 19727.8 25705.3 14337.6 49147.1 41808.7 4982.7 25076.8 25371.9 61937.0 79605.2 63574.3

PI 5.3 9.2 4.7 6.5 5.6 6.5 4.3 7.0 5.0 5.2 7.5 7.5

Extinction 28420 17880 28310 18450 49280 52370 8940 45380 24870 58680 72660 58790

Instability 50.4 46.8 41.4 29.0 36.3 29.8 29.7 42.1 39.4 28.8 30.0 32.2

Aliphatic 90.8 86.0 84.1 82.7 92.8 93.8 95.3 92.1 81.9 90.1 86.4 90.0

Hydrophathicity -0.245 -0.308 -0.248 -0.247 -0.258 -0.112 0.160 -0.258 -0.360 -0.200 -0.236 -0.344

 Human Identity 42.8 26.9 <10% 94.3 <10% 16.6 24.2 <10% <10% 26.7 19.6 56.6

VAR Identity 98.5 98.8 83.6 92.1 96.7 97.3 78.6 96.8 91.4 98.9 98.8 96.4

VAC Identity 99.0 100.0 83.6 \\ 97.4 98.7 95.2 98.6 94.1 98.9 99.1 97.8

CPV Identity 99.0 98.8 87.2 79.7 \\ 98.7 97.6 \\ 94.1 \\ \\ 97.8

Specific comment 4

Why only 12 proteins were selected as targets? Are all the 12 key proteins druggable? The authors need to verify this information as not all proteins present highly druggable surfaces for docking. The validity of the structures used for docking should be more adequately demonstrated.

Response 4

We are grateful to the reviewers for raising the issue of proteins druggable, and we will provide answers to each of the questions raised. Firstly, about the 12 target proteins selected: in the preliminary background research, this work extensively read all the articles (more than 1,000) on orthopox viruses since the first outbreak to seek for key targets that might inhibit orthopox viruses. During infection, a large number of proteins are involved in the process of virus invasion, replication, envelopment and release, and the virus can be effectively inhibited by inhibiting these key proteins. Based on literature research, 12 key proteins with important roles for the virus were finally searched and identified by summarizing the studies of various infection processes. The whole work does not involve screening proteins, but rather investigating all the targets that have been clearly shown to be potential in studies. The second problem is for the proteins druggable: the treatment of monkeypox is still in the exploratory stage, and there is no specific drug that can effectively deal with it. The 12 target proteins are all shown to be potentially druggable, and each protein is clearly described in section 3.2 of the article, explaining why it is a potential target for monkeypox treatment. In addition, all 12 proteins have clear docking sites, which were identified by previous researchers through various chemical experiments, not just theoretical descriptions. So, in response to the reviewer's question about the validity of the docking structure I think there is no problem.

Specific comment 5

Are any of the proteins intrinsically disordered? This is a very key question that needs answering. They are a fascinating class of proteins that lack a fixed or well-defined three-dimensional structure in their free state. Therefore, docking against such proteins are more challenging than what meets the eye. Therefore, physiochemical properties like isoelectric point (pI), molecular weight, aliphatic index, instability index, extinction co-efficient and residues accessibility of the 12 key proteins must be examined and provided in the manuscript. This data will aid experimental biologists to choose suitable proteins for experimental validation.

Overall, the proteins that are highly conserved, non-host homologous, highly druggable are usually considered very high value therapeutic targets. The proteins that ensure these features are likely to have a higher rate of success. Hence, I feel these key points need to be investigated by the authors before being considered for final consideration. The above recommended paper should aid the authors in this endeavor.

Response 5

We recognize in depth the issues raised by the reviewers. Among the 12 proteins we studied, there is no disordered protein structure. The three-dimensional structure of proteins serves as an important basis for understanding their biological functions and structure-based drug design. Using homologous mode-bonding technique, we obtained monkeypox target protein structures based on eight (A49R, C3L, H1L, E4R, C7L, A41L, D6L, and E13L) orthopoxviral (VAR, VAC, and CPV) proteins that were already available as templates. As for the key proteins (A22R, I7L, C19L and A50R), which a

---

## [Editor Report · Decision Letter 1]

21 May 2024

PONE-D-24-03917R1Identification of core therapeutic targets for Monkeypox virus and repurposing potential of drugs against them: an in-silico approachPLOS ONE

Dear Dr. Hu,

A concern has been raised of similar title and potential data overlapping with an article published by another group PMID: 37211001 (https://www.sciencedirect.com/science/article/pii/S0010482523004365?via%3Dihub).

Please address this concern explaining the similarities and differences between the two articles.

We look forward to receiving your revised manuscript.

Kind regards,

Mahmoud Kandeel

Academic Editor

PLOS ONE

Journal Requirements:

Additional Editor Comments:

A concern has been raised of similar title and potential data overlapping with an article published by another group PMID: 37211001 (https://www.sciencedirect.com/science/article/pii/S0010482523004365?via%3Dihub).

Please address this concern explaining the similarities and differences between the two articles.

---

## [Author Response · Author response to Decision Letter 1]

23 May 2024

Dear Editorial Office

Editor, PLOS ONE

Thank you very much for your email of Major decision, regarding our manuscript: Identifying the core therapeutic targets of monkeypox virus and the repurposing potential of orthopox inhibitors: an in-silico approach (Manuscript Number: PONE-D-24-03917R1).

It's good to hear from you, and we're responding to the questions mentioned by Editor. First of all, our article is duplicated with another article title. When the reviewer gave his comments, he suggested us to change the title several times and gave us this title. At that time, we thought the title was suitable for our article and applied it, forgetting to check whether it was a duplicate of another article's title. We will revise the title of the article appropriately. Secondly, the editor mentioned that the article has similarity in content. Both articles were used as the core therapeutic target of monkeypox protein, and it is necessary to understand the story behind monkeypox. In addition, the focus of the two articles is different, our article mainly focuses on the important targets and potential inhibitors of monkeypox to screen possible drugs and speculate other targets, while the other article focuses on the connection between specific targets and inhibitors. The originality of the article is a must for every researcher, and we guarantee that there is no plagiarism or falsification in the writing and experimental process of this article.

We look forward to receiving your revised manuscript.

Kind regards,

Dr. Hu

---

## [Editor Report · Decision Letter 2]

24 May 2024

PONE-D-24-03917R2Identifying the core therapeutic targets of monkeypox virus and the repurposing potential of orthopox inhibitors: an in-silico approachPLOS ONE

Dear Dr. Hu,

Thank you for submitting your manuscript to PLOS ONE. After careful consideration, we feel that it has merit but does not fully meet PLOS ONE’s publication criteria as it currently stands. Therefore, we invite you to submit a revised version of the manuscript that addresses the points raised during the review process.

We look forward to receiving your revised manuscript.

Kind regards,

Mahmoud Kandeel

Academic Editor

PLOS ONE

Journal Requirements:

Additional Editor Comments:

Please rewrite the title of the manuscript, to avoid overlapping and similarity with the previously published article. The title must clearly express the content of your manuscript.

---

## [Author Response · Author response to Decision Letter 2]

24 May 2024

Prof. Jianping Hu, PhD

Chengdu University 

610106 Chengdu, China

E-mail: hjpcdu@163.com

Chengdu, May 24th, 2024

Dear Editorial Office

Editor, PLOS ONE

Thank you very much for your email of Major decision, regarding our manuscript Development of mechanism‐based inhibitor against Monkeypox virus: A WEB to predict the binding patterns (Manuscript Number: PONE-D-24-03917R1).

We would like to thank wholeheartedly the referees for their valuable and helpful comments about our manuscript. Thanks to the reviewers for their recognition of the research objectives, rationale and methods of the article, as well as confirming the experimental data of the article. Suggestions on the figures and structure of the article will also be listened to and improved upon. Our replies to reviewers’ comments are enclosed below and in the revised version of the manuscript, we marked two parts in red: the suggestions given by the reviewers and the contents modified by the second reading of the article. We hope that our revised manuscript is now suitable for publication in PLOS ONE.

Yours sincerely,

Dr. Jianping Hu

E-mail: hjpcdu@163.com

Reply to Editor

Thank you to the editors for their advice on article acceptance. We have reconfirmed PLOS-ONE's requirements for articles as well as shared code to ensure that the submitted manuscripts are in order. Second, the funder had no role in the study design, data collection and analysis, publication decision, or preparation of the manuscript. Additionally, the editor mentioned a data sharing program, and we accept the editor's suggestion that we can accept sharing of data prior to article acceptance. The editor also raised the issue of copyright, and we were acutely aware of the importance of copyright, so we corrected the article and removed Figure s.17 Finally, for the supporting material we made a separate file and made sure that each information file had a legend in the manuscript following the list of references.

Reply to Reviewer 1

In this study the authors claim to have developed a monkeypox core protein library and a library of inhibitors targeting these core proteins. The authors have in addition built a docking server that predicts the binding modes between the protein and the substrate to understand the molecular dialogues between these entities. Currently there are hardly such servers dedicated for monkeypox virus. Therefore, such an endeavor is greatly welcome. Although the paper appears to be sound overall, I have some key recommendations that that I think can be incorporated before the paper can be accepted for publication.

Specific comment 1

In the methodology section method 2.1. Open reading frame (ORF), the ORF needs to be defined more concretely. I advise the authors to have a look at the results section (3.1 The meaning of the open reading frame (ORF) and core proteins in this study) of the paper titled “ Identification of core therapeutic targets for Monkeypox virus and repurposing potential of drugs against them: An in silico approach ”for better clarity.

Response 1

I appreciate the editor's comments and I have taken the time to check out the full article. The Open Reading Frame (ORF) serves as the beginning of the article and has an important role in the subsequent work. In the Methods section we considered the length of the article, so we simply explained the ORF. In response to the editor's suggestion, we have explained this section in more detail, and the specific corrections are as follows:

ORF is a fundamental concept in molecular bioinformatics, it is the normal nucleotide sequence of a structural gene, the reading frame from the start codon to the stop codon can encode a complete polypeptide chain, between which there is no stop codon that interrupts the translation. The term open refers to the "open" region of an intact gene used for protein translation, and reading frame refers to one of the six possibilities for translation of a double-stranded gene sequence into amino acids. In molecular biology, the detection of ORFs is an important step in the discovery of specific protein-coding genes in the genome sequence, and can serve as an indicator of a potential protein-coding gene. In our work, the acquisition of ORF mainly includes literature reading and database research.

In addition to the suggestions for ORFs, the editors also made a suggestion to change the title of the article to "Identification of core therapeutic targets for Monkeypox virus and repurposing potential of drugs against They: An in-silico approach", which is more in accordance with the content of the article. In our very first title, we targeted the server, ignoring the fact that the article studies more on monkeypox virus and core proteins. Seeing the editor's suggestion, we appreciate being able to identify this issue and also think that changing the title would benefit the article more.

Specific comment 2

Similarly, the term core protein creates a lot of confusion. From the view of a genomics researcher core protein is more closely associated with proteins that are highly conserved across all the genomes. However, from the point of view of a classical virologist core protein usually means the structural protein embedded on the outer membranes that give a virus its definitive shape. Therefore, the authors should clearly mention this in the manuscript as readers will comprise of people from multiple disciplines. Again, I advise the authors to have a look at the results section (3.1 The meaning of the open reading frame (ORF) and core proteins in this study) of the paper titled “Identification of core therapeutic targets for Monkeypox virus and repurposing potential of drugs against them: An in-silico approach” for better clarity.

Response 2

We thank the reviewers for this suggestion, and we do recognize the reviewers' suggestion that the core proteins were improperly worded. In our work, core proteins are used to denote potential proteins with the potential to inhibit monkeypox virus, and do not refer to structural proteins or highly conserved proteins as mentioned by the reviewer. In order to avoid misunderstanding, we have corrected all references to core proteins in the article and replaced the original core proteins with potential targets. The following are some of the corrections:

A new round of monkeypox virus has emerged in the United Kingdom since July 2022 and rapidly swept the world. Currently, despite numerous research groups are studying this virus and seeking effective treatments, the information on the open reading frame, inhibitors, and potential targets of monkeypox has not been updated in time, and the comprehension of monkeypox target ligand interactions remains a key challenge. Here, we first summarized and improved the open reading frame information of monkeypox, constructed the monkeypox inhibitor library and potential targets library by database research as well as literature search, combined with advanced protein modeling technologies (Sequence-based and AI algorithms-based homology modeling). In addition, we build monkeypox virus Docking Server, a web server to predict the binding mode between targets and substrate. The open reading frame information, monkeypox inhibitor library, and monkeypox potential targets library are used as the initial files for server docking, providing free interactive tools for predicting ligand interactions of monkeypox targets, potential drug screening, and potential targets search. In addition, the update of the three databases can also effectively promote the study of monkeypox drug inhibition mechanism and provide theoretical guidance for the development of drugs for monkeypox.

In this issue, the reviewer again proposed a change in the title of the article. In the previous comment, we clearly stated that we recognized the reviewer's title and explained the reason for the title in the first place. Again, we thank the reviewers for their attention to this issue.

Specific comment 3

Shared host homology is one of the biggest reasons for the failure of a drug from reaching the bed from the bench. This often creates adverse reactions by to off-targeting the host proteins instead. As a consequence, drugs often fail during clinical trials. Therefore, the authors must verify that the 12 key proteins do not share homology with human proteins beyond an allowable threshold (usually e-value of 0.0001 against the entire set of human proteome dataset (taxid: 9606) is considered a good threshold). The authors must provide this information in the manuscript.

Response 3

This question raised by the reviewer is very interesting and we have addressed it with a detailed explanation. Shared host homology as an important issue that must be considered for subsequent drug development, if the homology is too high it will lead to poor drug selectivity, and attacking the viral cells while also harming the host cells to a certain extent. In our article, this issue was neglected, so in the revised manuscript, for both host and viral cells we added this part. This includes calculation of physicochemical information of 12 proteins, sequence similarity with human-derived proteins, VAC, VAR and CPV, detailing the host homology issue.

The three-dimensional structure of proteins is an important basis for understanding their biological functions and for structure-based drug design [104]. With the rapid development of structural biology, the speed of measuring protein 3D structures using methods such as NMR, X-Ray or cryoelectronic microscopy (Cryo-EM) crystallography has been greatly improved [39, 105]. However, the 3D structures of some proteins are still difficult to determine due to many reasons, such as excessive molecular weight or difficulty in crystallization. Before modeling the protein structures, we first performed a series of physicochemical property analyses on 12 potential target sequences, including Weight, PI, Extinction, Instability, Aliphatic and Hydrophathicity, and compared the sequences of monkeypox with those of human origin and other orthopox viruses (table s1). It was found that the structural domains of the proteins were stabilized to a certain extent and could be used for subsequent studies; The sequence similarity of all proteins except D6L was less than 60% to human, which avoided the side effects of the drug caused by host homology. D6L, as an interleukin-18 binding protein homologous to the host, functionally affirmed its high sequence similarity to human proteins, a characteristic that also increased the difficulty of developing drugs against it; finally, the sequence similarity between monkeypox virus and orthopoxvirus was higher than 78%, proving the existence of commonality between them, and providing an opportunity to seek possible orthopoxvirus-based drugs for the subsequent studies. This lays a theoretical foundation for the search of possible monkeypox virus drugs based on orthopox viruses.

For the key proteins of monkeypox, we found that eight (A49R, C3L, H1L, E4R, C7L, A41L, D6L and E13L) could be obtained from orthopox viruses (VAR, VAC and MPV) modeled with high sequence similarity. As for the key proteins (A22R, I7L, C19L and A50R), which are not present in any of the orthopoxvirus genera, we chose AlphaFold2, which is only one atomic width away from the true structure for most of the predicted protein structures, to predict them. Notably that based on the multifaceted superiority, AlphaFold2 reaches the level of prediction observed by sophisticated instruments such as cryoelectron microscopy, which can effectively guarantee the accuracy of the predicted structures (Fig.2) [106]. Hence, we constructed a database of orthopoxvirus proteins with a capacity of 45.

Table s1 Physicochemical properties of 12 potential target protein sequences and sequence similarity to human and orthopoxvirus

 A49R H1L C7L D6L A22R C19L C3L E4R A41L E13L I7L A50R

Weight 23289.7 19727.8 25705.3 14337.6 49147.1 41808.7 4982.7 25076.8 25371.9 61937.0 79605.2 63574.3

PI 5.3 9.2 4.7 6.5 5.6 6.5 4.3 7.0 5.0 5.2 7.5 7.5

Extinction 28420 17880 28310 18450 49280 52370 8940 45380 24870 58680 72660 58790

Instability 50.4 46.8 41.4 29.0 36.3 29.8 29.7 42.1 39.4 28.8 30.0 32.2

Aliphatic 90.8 86.0 84.1 82.7 92.8 93.8 95.3 92.1 81.9 90.1 86.4 90.0

Hydrophathicity -0.245 -0.308 -0.248 -0.247 -0.258 -0.112 0.160 -0.258 -0.360 -0.200 -0.236 -0.344

 Human Identity 42.8 26.9 <10% 94.3 <10% 16.6 24.2 <10% <10% 26.7 19.6 56.6

VAR Identity 98.5 98.8 83.6 92.1 96.7 97.3 78.6 96.8 91.4 98.9 98.8 96.4

VAC Identity 99.0 100.0 83.6 \\ 97.4 98.7 95.2 98.6 94.1 98.9 99.1 97.8

CPV Identity 99.0 98.8 87.2 79.7 \\ 98.7 97.6 \\ 94.1 \\ \\ 97.8

Specific comment 4

Why only 12 proteins were selected as targets? Are all the 12 key proteins druggable? The authors need to verify this information as not all proteins present highly druggable surfaces for docking. The validity of the structures used for docking should be more adequately demonstrated.

Response 4

We are grateful to the reviewers for raising the issue of proteins druggable, and we will provide answers to each of the questions raised. Firstly, about the 12 target proteins selected: in the preliminary background research, this work extensively read all the articles (more than 1,000) on orthopox viruses since the first outbreak to seek for key targets that might inhibit orthopox viruses. During infection, a large number of proteins are involved in the process of virus invasion, replication, envelopment and release, and the virus can be effectively inhibited by inhibiting these key proteins. Based on literature research, 12 key proteins with important roles for the virus were finally searched and identified by summarizing the studies of various infection processes. The whole work does not involve screening proteins, but rather investigating all the targets that have been clearly shown to be potential in studies. The second problem is for the proteins druggable: the treatment of monkeypox is still in the exploratory stage, and there is no specific drug that can effectively deal with it. The 12 target proteins are all shown to be potentially druggable, and each protein is clearly described in section 3.2 of the article, explaining why it is a potential target for monkeypox treatment. In addition, all 12 proteins have clear docking sites, which were identified by previous researchers through various chemical experiments, not just theoretical descriptions. So, in response to the reviewer's question about the validity of the docking structure I think there is no problem.

Specific comment 5

Are any of the proteins intrinsically disordered? This is a very key question that needs answering. They are a fascinating class of proteins that lack a fixed or well-defined three-dimensional structure in their free state. Therefore, docking against such proteins are more challenging than what meets the eye. Therefore, physiochemical properties like isoelectric point (pI), molecular weight, aliphatic index, instability index, extinction co-efficient and residues accessibility of the 12 key proteins must be examined and provided in the manuscript. This data will aid experimental biologists to choose suitable proteins for experimental validation.

Overall, the proteins that are highly conserved, non-host homologous, highly druggable are usually considered very high value therapeutic targets. The proteins that ensure these features are likely to have a higher rate of success. Hence, I feel these key points need to be investigated by the authors before being considered for final consideration. The above recommended paper should aid the authors in this endeavor.

Response 5

We recognize in depth the issues raised by the reviewers. Among the 12 proteins we studied, there is no disordered protein structure. The three-dimensional structure of proteins serves as an important basis for understanding their biological functions and structure-based drug design. Using homologous mode-bonding technique, we obtained monkeypox target protein structures based on eight (A49R, C3L, H1L, E4R, C7L, A41L, D6L, and E13L) orthopoxviral (VAR, VAC, and CPV) proteins that were already available as templates. As for the key proteins (A22R, I7L, C19L and A50R), which are

---

## [Editor Report · Decision Letter 3]

31 May 2024

PONE-D-24-03917R3Identifying the core therapeutic targets of monkeypox virus and the repurposing potential of orthopox inhibitors: an in-silico approachPLOS ONE

Dear Dr. Hu,

Thank you for submitting your manuscript to PLOS ONE. After careful consideration, we feel that it has merit but does not fully meet PLOS ONE’s publication criteria as it currently stands. Therefore, we invite you to submit a revised version of the manuscript that addresses the points raised during the review process.

**ACADEMIC EDITOR: **Please rewrite the title of the manuscript, to avoid overlapping and similarity with the previously published article. The title must clearly express the content of your manuscript.

We look forward to receiving your revised manuscript.

Kind regards,

Mahmoud Kandeel

Academic Editor

PLOS ONE
---

## [Author Response · Author response to Decision Letter 3]

1 Jun 2024

Prof. Jianping Hu, PhD

Chengdu University 

610106 Chengdu, China

E-mail: hjpcdu@163.com

Chengdu, Jun 1th, 2024

Dear Editorial Office

Editor, PLOS ONE

Thank you very much for your email of Major decision, regarding our manuscript Core therapeutic targets identification for monkeypox virus and repurposing of orthopox inhibitors：an in-silico approach (Manuscript Number: PONE-D-24-03917R1).

We would like to thank wholeheartedly the referees for their valuable and helpful comments about our manuscript. Thanks to the reviewers for their recognition of the research objectives, rationale and methods of the article, as well as confirming the experimental data of the article. Suggestions on the figures and structure of the article will also be listened to and improved upon. Our replies to reviewers’ comments are enclosed below and in the revised version of the manuscript, we marked two parts in red: the suggestions given by the reviewers and the contents modified by the second reading of the article. We hope that our revised manuscript is now suitable for publication in PLOS ONE.

Yours sincerely,

Dr. Jianping Hu

E-mail: hjpcdu@163.com

Reply to Editor

Thank you to the editors for their advice on article acceptance. We have reconfirmed PLOS-ONE's requirements for articles as well as shared code to ensure that the submitted manuscripts are in order. Second, the funder had no role in the study design, data collection and analysis, publication decision, or preparation of the manuscript. Additionally, the editor mentioned a data sharing program, and we accept the editor's suggestion that we can accept sharing of data prior to article acceptance. The editor also raised the issue of copyright, and we were acutely aware of the importance of copyright, so we corrected the article and removed Figure s.17. Finally, for the supporting material we made a separate file and made sure that each information file had a legend in the manuscript following the list of references.

Reply to Reviewer 1

In this study the authors claim to have developed a monkeypox core protein library and a library of inhibitors targeting these core proteins. The authors have in addition built a docking server that predicts the binding modes between the protein and the substrate to understand the molecular dialogues between these entities. Currently there are hardly such servers dedicated for monkeypox virus. Therefore, such an endeavor is greatly welcome. Although the paper appears to be sound overall, I have some key recommendations that that I think can be incorporated before the paper can be accepted for publication.

Specific comment 1

In the methodology section method 2.1. Open reading frame (ORF), the ORF needs to be defined more concretely. I advise the authors to have a look at the results section (3.1 The meaning of the open reading frame (ORF) and core proteins in this study) of the paper titled “ Identification of core therapeutic targets for Monkeypox virus and repurposing potential of drugs against them: An in silico approach ”for better clarity.

Response 1

I appreciate the editor's comments and I have taken the time to check out the full article. The Open Reading Frame (ORF) serves as the beginning of the article and has an important role in the subsequent work. In the Methods section we considered the length of the article, so we simply explained the ORF. In response to the editor's suggestion, we have explained this section in more detail, and the specific corrections are as follows:

ORF is a fundamental concept in molecular bioinformatics, it is the normal nucleotide sequence of a structural gene, the reading frame from the start codon to the stop codon can encode a complete polypeptide chain, between which there is no stop codon that interrupts the translation. The term open refers to the "open" region of an intact gene used for protein translation, and reading frame refers to one of the six possibilities for translation of a double-stranded gene sequence into amino acids. In molecular biology, the detection of ORFs is an important step in the discovery of specific protein-coding genes in the genome sequence, and can serve as an indicator of a potential protein-coding gene. In our work, the acquisition of ORF mainly includes literature reading and database research.

In addition to the suggestions for ORFs, the editors also made a suggestion to change the title of the article to " Core therapeutic targets identification for monkeypox virus and repurposing of orthopox inhibitors: an in-silico approach", which is more in accordance with the content of the article. In our very first title, we targeted the server, ignoring the fact that the article studies more on monkeypox virus and core proteins. Seeing the editor's suggestion, we appreciate being able to identify this issue and also think that changing the title would benefit the article more.

Specific comment 2

Similarly, the term core protein creates a lot of confusion. From the view of a genomics researcher core protein is more closely associated with proteins that are highly conserved across all the genomes. However, from the point of view of a classical virologist core protein usually means the structural protein embedded on the outer membranes that give a virus its definitive shape. Therefore, the authors should clearly mention this in the manuscript as readers will comprise of people from multiple disciplines. Again, I advise the authors to have a look at the results section (3.1). The meaning of the open reading frame (ORF) and core proteins in this study) of the paper titled “Identification of core therapeutic targets for Monkeypox virus and repurposing potential of drugs against them: An in-silico approach” for better clarity.

Response 2

We thank the reviewers for this suggestion, and we do recognize the reviewers' suggestion that the core proteins were improperly worded. In our work, core proteins are used to denote potential proteins with the potential to inhibit monkeypox virus, and do not refer to structural proteins or highly conserved proteins as mentioned by the reviewer. In order to avoid misunderstanding, we have corrected all references to core proteins in the article and replaced the original core proteins with potential targets. The following are some of the corrections:

A new round of monkeypox virus has emerged in the United Kingdom since July 2022 and rapidly swept the world. Currently, despite numerous research groups are studying this virus and seeking effective treatments, the information on the open reading frame, inhibitors, and potential targets of monkeypox has not been updated in time, and the comprehension of monkeypox target ligand interactions remains a key challenge. Here, we first summarized and improved the open reading frame information of monkeypox, constructed the monkeypox inhibitor library and potential targets library by database research as well as literature search, combined with advanced protein modeling technologies (Sequence-based and AI algorithms-based homology modeling). In addition, we build monkeypox virus Docking Server, a web server to predict the binding mode between targets and substrate. The open reading frame information, monkeypox inhibitor library, and monkeypox potential targets library are used as the initial files for server docking, providing free interactive tools for predicting ligand interactions of monkeypox targets, potential drug screening, and potential targets search. In addition, the update of the three databases can also effectively promote the study of monkeypox drug inhibition mechanism and provide theoretical guidance for the development of drugs for monkeypox.

In this issue, the reviewer again proposed a change in the title of the article. In the previous comment, we clearly stated that we recognized the reviewer's title and explained the reason for the title in the first place. Again, we thank the reviewers for their attention to this issue.

Specific comment 3

Shared host homology is one of the biggest reasons for the failure of a drug from reaching the bed from the bench. This often creates adverse reactions by to off-targeting the host proteins instead. As a consequence, drugs often fail during clinical trials. Therefore, the authors must verify that the 12 key proteins do not share homology with human proteins beyond an allowable threshold (usually e-value of 0.0001 against the entire set of human proteome dataset (taxid: 9606) is considered a good threshold). The authors must provide this information in the manuscript.

Response 3

This question raised by the reviewer is very interesting and we have addressed it with a detailed explanation. Shared host homology as an important issue that must be considered for subsequent drug development, if the homology is too high it will lead to poor drug selectivity, and attacking the viral cells while also harming the host cells to a certain extent. In our article, this issue was neglected, so in the revised manuscript, for both host and viral cells we added this part. This includes calculation of physicochemical information of 12 proteins, sequence similarity with human-derived proteins, VAC, VAR and CPV, detailing the host homology issue.

The three-dimensional structure of proteins is an important basis for understanding their biological functions and for structure-based drug design [104]. With the rapid development of structural biology, the speed of measuring protein 3D structures using methods such as NMR, X-Ray or cryoelectronic microscopy (Cryo-EM) crystallography has been greatly improved [39, 105]. However, the 3D structures of some proteins are still difficult to determine due to many reasons, such as excessive molecular weight or difficulty in crystallization. Before modeling the protein structures, we first performed a series of physicochemical property analyses on 12 potential target sequences, including Weight, PI, Extinction, Instability, Aliphatic and Hydrophathicity, and compared the sequences of monkeypox with those of human origin and other orthopox viruses (table s1). It was found that the structural domains of the proteins were stabilized to a certain extent and could be used for subsequent studies; The sequence similarity of all proteins except D6L was less than 60% to human, which avoided the side effects of the drug caused by host homology. D6L, as an interleukin-18 binding protein homologous to the host, functionally affirmed its high sequence similarity to human proteins, a characteristic that also increased the difficulty of developing drugs against it; finally, the sequence similarity between monkeypox virus and orthopoxvirus was higher than 78%, proving the existence of commonality between them, and providing an opportunity to seek possible orthopoxvirus-based drugs for the subsequent studies. This lays a theoretical foundation for the search of possible monkeypox virus drugs based on orthopox viruses.

For the key proteins of monkeypox, we found that eight (A49R, C3L, H1L, E4R, C7L, A41L, D6L and E13L) could be obtained from orthopox viruses (VAR, VAC and MPV) modeled with high sequence similarity. As for the key proteins (A22R, I7L, C19L and A50R), which are not present in any of the orthopoxvirus genera, we chose AlphaFold2, which is only one atomic width away from the true structure for most of the predicted protein structures, to predict them. Notably that based on the multifaceted superiority, AlphaFold2 reaches the level of prediction observed by sophisticated instruments such as cryoelectron microscopy, which can effectively guarantee the accuracy of the predicted structures (Fig.2) [106]. Hence, we constructed a database of orthopoxvirus proteins with a capacity of 45.

Table s1 Physicochemical properties of 12 potential target protein sequences and sequence similarity to human and orthopoxvirus

 A49R H1L C7L D6L A22R C19L C3L E4R A41L E13L I7L A50R

Weight 23289.7 19727.8 25705.3 14337.6 49147.1 41808.7 4982.7 25076.8 25371.9 61937.0 79605.2 63574.3

PI 5.3 9.2 4.7 6.5 5.6 6.5 4.3 7.0 5.0 5.2 7.5 7.5

Extinction 28420 17880 28310 18450 49280 52370 8940 45380 24870 58680 72660 58790

Instability 50.4 46.8 41.4 29.0 36.3 29.8 29.7 42.1 39.4 28.8 30.0 32.2

Aliphatic 90.8 86.0 84.1 82.7 92.8 93.8 95.3 92.1 81.9 90.1 86.4 90.0

Hydrophathicity -0.245 -0.308 -0.248 -0.247 -0.258 -0.112 0.160 -0.258 -0.360 -0.200 -0.236 -0.344

 Human Identity 42.8 26.9 <10% 94.3 <10% 16.6 24.2 <10% <10% 26.7 19.6 56.6

VAR Identity 98.5 98.8 83.6 92.1 96.7 97.3 78.6 96.8 91.4 98.9 98.8 96.4

VAC Identity 99.0 100.0 83.6 \\ 97.4 98.7 95.2 98.6 94.1 98.9 99.1 97.8

CPV Identity 99.0 98.8 87.2 79.7 \\ 98.7 97.6 \\ 94.1 \\ \\ 97.8

Specific comment 4

Why only 12 proteins were selected as targets? Are all the 12 key proteins druggable? The authors need to verify this information as not all proteins present highly druggable surfaces for docking. The validity of the structures used for docking should be more adequately demonstrated.

Response 4

We are grateful to the reviewers for raising the issue of proteins druggable, and we will provide answers to each of the questions raised. Firstly, about the 12 target proteins selected: in the preliminary background research, this work extensively read all the articles (more than 1,000) on orthopox viruses since the first outbreak to seek for key targets that might inhibit orthopox viruses. During infection, a large number of proteins are involved in the process of virus invasion, replication, envelopment and release, and the virus can be effectively inhibited by inhibiting these key proteins. Based on literature research, 12 key proteins with important roles for the virus were finally searched and identified by summarizing the studies of various infection processes. The whole work does not involve screening proteins, but rather investigating all the targets that have been clearly shown to be potential in studies. The second problem is for the proteins druggable: the treatment of monkeypox is still in the exploratory stage, and there is no specific drug that can effectively deal with it. The 12 target proteins are all shown to be potentially druggable, and each protein is clearly described in section 3.2 of the article, explaining why it is a potential target for monkeypox treatment. In addition, all 12 proteins have clear docking sites, which were identified by previous researchers through various chemical experiments, not just theoretical descriptions. So, in response to the reviewer's question about the validity of the docking structure I think there is no problem.

Specific comment 5

Are any of the proteins intrinsically disordered? This is a very key question that needs answering. They are a fascinating class of proteins that lack a fixed or well-defined three-dimensional structure in their free state. Therefore, docking against such proteins are more challenging than what meets the eye. Therefore, physiochemical properties like isoelectric point (pI), molecular weight, aliphatic index, instability index, extinction co-efficient and residues accessibility of the 12 key proteins must be examined and provided in the manuscript. This data will aid experimental biologists to choose suitable proteins for experimental validation.

Overall, the proteins that are highly conserved, non-host homologous, highly druggable are usually considered very high value therapeutic targets. The proteins that ensure these features are likely to have a higher rate of success. Hence, I feel these key points need to be investigated by the authors before being considered for final consideration. The above recommended paper should aid the authors in this endeavor.

Response 5

We recognize in depth the issues raised by the reviewers. Among the 12 proteins we studied, there is no disordered protein structure. The three-dimensional structure of proteins serves as an important basis for understanding their biological functions and structure-based drug design. Using homologous mode-bonding technique, we obtained monkeypox target protein structures based on eight (A49R, C3L, H1L, E4R, C7L, A41L, D6L, and E13L) orthopoxviral (VAR, VAC, and CPV) proteins that were already available as templates. As for the key proteins (A22R, I7L, C19L and A50R), w

---

## [Editor Report · Decision Letter 4]

10 Jun 2024

PONE-D-24-03917R4Core therapeutic targets identification for monkeypox virus and repurposing of orthopox inhibitors: an in-silico approachPLOS ONE

Dear Dr. Hu,

Thank you for submitting your manuscript to PLOS ONE. After careful consideration, we feel that it has merit but does not fully meet PLOS ONE’s publication criteria as it currently stands. Therefore, we invite you to submit a revised version of the manuscript that addresses the points raised during the review process. Please submit your revised manuscript by Jul 25 2024 11:59PM. If you will need more time than this to complete your revisions, please reply to this message or contact the journal office at plosone@plos.org. Please include the following items when submitting your revised manuscript:A rebuttal letter that responds to each point raised by the academic editor and reviewer(s). You should upload this letter as a separate file labeled 'Response to Reviewers'.A marked-up copy of your manuscript that highlights changes made to the original version. You should upload this as a separate file labeled 'Revised Manuscript with Track Changes'.An unmarked version of your revised paper without tracked changes. You should upload this as a separate file labeled 'Manuscript'.If applicable, we recommend that you deposit your laboratory protocols in protocols.io to enhance the reproducibility of your results. Protocols.io assigns your protocol its own identifier (DOI) so that it can be cited independently in the future. For instructions see: https://journals.plos.org/plosone/s/submission-guidelines#loc-laboratory-protocols. Additionally, PLOS ONE offers an option for publishing peer-reviewed Lab Protocol articles, which describe protocols hosted on protocols.io. Read more information on sharing protocols at https://plos.org/protocols?utm_medium=editorial-email&utm_source=authorletters&utm_campaign=protocols.

We look forward to receiving your revised manuscript.

Kind regards,

Mahmoud Kandeel

Academic Editor

PLOS ONE

Journal Requirements:

**Additional Editor Comments:**

Dear dr Hu

Please resubmit your article using the initial title used during the first submission

Best regards

---

## [Decision Letter · Decision Letter 5]

31 Jul 2024

PONE-D-24-03917R5Development of mechanism‐based inhibitor against Monkeypox virus: A WEB to predict the binding patternsPLOS ONE

Dear Dr. Hu,

Thank you for submitting your manuscript to PLOS ONE. After careful consideration, we feel that it has merit but does not fully meet PLOS ONE’s publication criteria as it currently stands. Therefore, we invite you to submit a revised version of the manuscript that addresses the points raised during the review process.

We look forward to receiving your revised manuscript.

Kind regards,

Milad Zandi, Ph.D.

Academic Editor

PLOS ONE

Journal Requirements:

Reviewers' comments:

Reviewer's Responses to Questions

**Comments to the Author**

1. If the authors have adequately addressed your comments raised in a previous round of review and you feel that this manuscript is now acceptable for publication, you may indicate that here to bypass the “Comments to the Author” section, enter your conflict of interest statement in the “Confidential to Editor” section, and submit your "Accept" recommendation.

Reviewer #3: All comments have been addressed

Reviewer #4: (No Response)

2. Is the manuscript technically sound, and do the data support the conclusions?

Reviewer #3: Yes

Reviewer #4: Yes

3. Has the statistical analysis been performed appropriately and rigorously? 

Reviewer #3: Yes

Reviewer #4: Yes

4. Have the authors made all data underlying the findings in their manuscript fully available?

Reviewer #3: Yes

Reviewer #4: Yes

5. Is the manuscript presented in an intelligible fashion and written in standard English?

Reviewer #3: Yes

Reviewer #4: Yes

6. Review Comments to the Author

**Reviewer #3:** The article provides a detailed introduction on monkeypox, summarises and refines the open reading frame information of monkeypox, and constructs a model of monkeypox target-ligand interactions. In addition, a monkeypox virus docking server, which is a web server for predicting the binding mode of targets and substrates, was constructed, which is conducive to the further advancement of monkeypox virus drug screening and target identification. The amount of data in the article is sufficient, the analysis is reasonable and there are no ethical issues, I recommend plos one to receive the manuscript.

**Reviewer #4:** In this study the authors claim to have developed a monkeypox core protein library and a library of inhibitors targeting these core proteins. The authors have in addition built a docking server that predicts the binding modes between the protein and the substrate to understand the molecular dialogues between these entities. Currently there are hardly such servers dedicated for monkeypox virus. Therefore, such an endeavor is greatly welcome. While the paper as a whole looks good, I have a few tips that I think need to be explained before the paper is accepted for publication.

1.Why only 12 proteins were selected as targets? Are all the 12 key proteins druggable? The authors need to verify this information as not all proteins present highly druggable surfaces for docking. The validity of the structures used for docking should be more adequately demonstrated.

2.Given that a significant portion of the protein crystal structures remains unresolved, the authors generated homology models to address this limitation. How did the authors predict the orthosteric sites of the protein targets? Was a validation conducted against known inhibitors to assess the accuracy of the predicted orthosteric sites?

3.The manuscript title can also be improved. It does not resonate with the work done or the core feature of the server.

7. PLOS authors have the option to publish the peer review history of their article (what does this mean?). If published, this will include your full peer review and any attached files.

Reviewer #3: No

Reviewer #4: No

---

## [Author Response · Author response to Decision Letter 5]

3 Aug 2024

Prof. Jianping Hu, PhD

Chengdu University 

610106 Chengdu, China

E-mail: hjpcdu@163.com

Chengdu, July 31th, 2024

Dear Editorial Office

Editor, PLOS ONE

Thank you very much for your email of Major decision, regarding our manuscript Development of mechanism‐based inhibitor against Monkeypox virus: A WEB to predict the binding patterns (Manuscript Number: PONE-D-24-03917R1).

We would like to thank wholeheartedly the referees for their valuable and helpful comments about our manuscript. Thanks to the reviewers for their recognition of the research objectives, rationale and methods of the article, as well as confirming the experimental data of the article. Suggestions on the figures and structure of the article will also be listened to and improved upon. Our replies to reviewers’ comments are enclosed below and in the revised version of the manuscript, we marked two parts in red: the suggestions given by the reviewers and the contents modified by the second reading of the article. We hope that our revised manuscript is now suitable for publication in PLOS ONE.

Yours sincerely,

Dr. Jianping Hu

E-mail: hjpcdu@163.com

Reviewer #3:

The article provides a detailed introduction on monkeypox, summarises and refines the open reading frame information of monkeypox, and constructs a model of monkeypox target-ligand interactions. In addition, a monkeypox virus docking server, which is a web server for predicting the binding mode of targets and substrates, was constructed, which is conducive to the further advancement of monkeypox virus drug screening and target identification. The amount of data in the article is sufficient, the analysis is reasonable and there are no ethical issues, I recommend plos one to receive the manuscript.

Reviewer #4:

In this study the authors claim to have developed a monkeypox core protein library and a library of inhibitors targeting these core proteins. The authors have in addition built a docking server that predicts the binding modes between the protein and the substrate to understand the molecular dialogues between these entities. Currently there are hardly such servers dedicated for monkeypox virus. Therefore, such an endeavor is greatly welcome. While the paper as a whole looks good, I have a few tips that I think need to be explained before the paper is accepted for publication.

Specific comment 1

Why only 12 proteins were selected as targets? Are all the 12 key proteins druggable? The authors need to verify this information as not all proteins present highly druggable surfaces for docking. The validity of the structures used for docking should be more adequately demonstrated.

Response 1

We are grateful to the reviewers for raising the issue of proteins druggable, and we will provide answers to each of the questions raised. Firstly, about the 12 target proteins selected: in the preliminary background research, this work extensively read all the articles (more than 1,000) on orthopox viruses since the first outbreak to seek for key targets that might inhibit orthopox viruses. During infection, a large number of proteins are involved in the process of virus invasion, replication, envelopment and release, and the virus can be effectively inhibited by inhibiting these key proteins. Based on literature research, 12 key proteins with important roles for the virus were finally searched and identified by summarizing the studies of various infection processes. The whole work does not involve screening proteins, but rather investigating all the targets that have been clearly shown to be potential in studies. The second problem is for the proteins druggable: the treatment of monkeypox is still in the exploratory stage, and there is no specific drug that can effectively deal with it. The 12 target proteins are all shown to be potentially druggable, and each protein is clearly described in section 3.2 of the article, explaining why it is a potential target for monkeypox treatment. In addition, all 12 proteins have clear docking sites, which were identified by previous researchers through various chemical experiments, not just theoretical descriptions. So, in response to the reviewer's question about the validity of the docking structure I think there is no problem.

Specific comment 2

Given that a significant portion of the protein crystal structures remains unresolved, the authors generated homology models to address this limitation. How did the authors predict the orthosteric sites of the protein targets? Was a validation conducted against known inhibitors to assess the accuracy of the predicted orthosteric sites?

Response 2

Thanks to the reviewers for their professional advice. Proteins are important functional molecules in living organisms, and their structures determine their functions and interaction modes. Accurate three-dimensional structures of proteins are of key significance in revealing their biological functions, disease mechanisms, and drug development. In our work, the monkeypox virus protein structure was targeted mainly by template-based homology modal bonding and sequence-based modal bonding. Discovery Studio program (DS), as a professional molecular simulation software for life sciences, has been used in the characterization of proteins (including protein-protein interactions), homology modeling, molecular mechanics calculations and molecular dynamics simulations, structure-based drug design tools ( including ligand-protein interactions, novel drug design and molecular docking), small-molecule-based drug design tools (including quantitative conformational relationships, pharmacophore, database screening, ADMET), and design and analysis of combinatorial libraries. By using DS, a reasonable protein structure can be well obtained based on amino acid sequence mode bonds with certain sequence similarity, and this technique is also widely used and verified to be reasonable by major groups. For the sequence similarity content of protein structures, we have also made corresponding additions to demonstrate the high sequence similarity among orthopox viruses and the rationality of obtaining structures through homologous mold bonds. Here we give some literature on the use of the program to die bond protein structures in order to open the work: Studio D. Accelrys, 2008, 420; Evers A, Klebe G. Journal of medicinal chemistry, 2004, 47(22): 5381-5392; Duan H, Hu K, Zheng D, et al. International Journal of Biological Macromolecules, 2022, 223: 1562-1577. For protein structures without templates, we chose the AlphaFold2 technique, which has gained more recognition in recent years. As a deep machine learning method, AlphaFold2 achieves an average accuracy of 92.4 for predicting proteins from scratch, which is almost comparable to experimentally resolved structures. Using the AlphaFold2 technique, we selected the highest scoring structures as rational structures and used them for subsequent drug screening work. In addition, the reviewer asked how to predict the orthosteric site of the protein and whether to verify the site with inhibitors. We thank the reviewers for asking this question again, and we understand that the orthosteric site is the active site that exerts binding to the substrate. In the previous question we explained that all binding sites were obtained experimentally and not by molecular modeling techniques. For the active site validation problem, we also gave a detailed explanation in the third question and corrected the content of the main text. The docking results showed that more than 68% of the compounds in the small molecule library were successfully bound to the correct site, which proves the accuracy of the active site from the side.

Specific comment 3

The manuscript title can also be improved. It does not resonate with the work done or the core feature of the server.

Response 3

Regarding the manuscript title issue. We thank the reviewer for raising this issue, and we have also adjusted the title of the manuscript based on summarizing the full text, and the new title is entitled: Identification of core therapeutic targets for Monkeypox virus and repurposing potential of drugs: A WEB prediction approach.

---

## [Decision Letter · Decision Letter 6]

9 Oct 2024

确定猴痘病毒的核心治疗靶点并重新利用药物的潜力：一种WEB预测方法。

PONE-D-24-03917R6

Dear Dr. Hu,

We’re pleased to inform you that your manuscript has been judged scientifically suitable for publication and will be formally accepted for publication once it meets all outstanding technical requirements.

Kind regards,

Sheikh Arslan Sehgal, PhD

Academic Editor

PLOS ONE

Additional Editor Comments (optional):

Reviewers' comments:

Reviewer's Responses to Questions

**Comments to the Author**

1. If the authors have adequately addressed your comments raised in a previous round of review and you feel that this manuscript is now acceptable for publication, you may indicate that here to bypass the “Comments to the Author” section, enter your conflict of interest statement in the “Confidential to Editor” section, and submit your "Accept" recommendation.

Reviewer #3: All comments have been addressed

Reviewer #4: All comments have been addressed

2. Is the manuscript technically sound, and do the data support the conclusions?

Reviewer #3: Yes

Reviewer #4: Yes

3. Has the statistical analysis been performed appropriately and rigorously? 

Reviewer #3: Yes

Reviewer #4: N/A

4. Have the authors made all data underlying the findings in their manuscript fully available?

Reviewer #3: Yes

Reviewer #4: Yes

5. Is the manuscript presented in an intelligible fashion and written in standard English?

Reviewer #3: Yes

Reviewer #4: Yes

6. Review Comments to the Author

Reviewer #3: The author has solved my problem. Recommend your journal to receive the article in its current version.

Reviewer #4: All of my concerns have been addressed. I recommend publishing this manuscript in its current format.

7. PLOS authors have the option to publish the peer review history of their article (what does this mean?). If published, this will include your full peer review and any attached files.

Reviewer #3: No

Reviewer #4: **Yes: **Zhiwei Feng

---

## [Editor Report · Acceptance letter]

14 Oct 2024

PONE-D-24-03917R6 

PLOS ONE

Dear Dr. Hu, 

I'm pleased to inform you that your manuscript has been deemed suitable for publication in PLOS ONE. Congratulations! Your manuscript is now being handed over to our production team.

Kind regards, 

on behalf of

Dr Sheikh Arslan Sehgal 

Academic Editor

PLOS ONE